# Continuous Diffusion Model for Language Modeling

**Jaehyeong Jo**[1], **Sung Ju Hwang**[1,2]

KAIST[1], DeepAuto.ai[2]

{ harryjo97, sjhwang82 }@kaist.ac.kr

## Abstract

Diffusion models have emerged as a promising alternative to autoregressive models in modeling discrete categorical data. However, diffusion models that directly work on discrete data space fail to fully exploit the power of iterative refinement, as the signals are lost during transitions between discrete states. Existing continuous diffusion models for discrete data underperform compared to discrete methods, and the lack of a clear connection between the two approaches hinders the development of effective diffusion models for discrete data. In this work, we propose a continuous diffusion model for language modeling that incorporates the geometry of the underlying categorical distribution. We establish a connection between the discrete diffusion and continuous flow on the statistical manifold, and building on this analogy, introduce a simple diffusion process that generalizes existing discrete diffusion models. We further propose a simulation-free training framework based on radial symmetry, along with a simple technique to address the high dimensionality of the manifold. Comprehensive experiments on language modeling benchmarks and other modalities show that our method outperforms existing discrete diffusion models and approaches the performance of autoregressive models. The code is available at https://github.com/harryjo97/RDLM.

## 1 Introduction

Discrete diffusion models [2, 39] emerged as a promising competitor to autoregressive models for the generative modeling of discrete data. These models have demonstrated competitive performance on tasks such as language modeling [49, 52] and code generation [20]. Unlike autoregressive models that generate data sequentially, diffusion models generate the sequence in parallel, allowing for bidirectional controllable generation and faster sampling.

However, discrete diffusion models do not fully harness the power of iterative refinement, which is the key to generative modeling of continuous data such as image synthesis [19, 48] and video generation [5, 46]. In these models, the forward process progressively corrupts data through stochastic jumps between discrete states, modeled as a Markov chain. Denoising is achieved through transitions between these discrete states, which results in the loss of informative signals during refinement. Hence, discrete diffusion models often exhibit limited generative performance and reduced controllability.

Several efforts have been made to adapt continuous diffusion models for discrete data, motivated by their advantages in controllability [26], efficient sampling [40, 41], optimized design choices [10, 34], and the potential to unify different modalities [35, 56]. However, their performance often significantly lags behind that of discrete diffusion models. Early methods [24, 36] extended image diffusion models to discrete domains by applying unconstrained continuous relaxation. Other approaches [3, 54] project discrete data onto the probability simplex using the Dirichlet distribution as its prior over categorical distributions, but often fail to capture complex patterns. Recent works [12, 15] apply flow matching on the statistical manifold to learn categorical distributions, but these methods are limited to short sequences and small vocabularies. In particular, the connection between discrete and continuous diffusion remains poorly understood, hindering the development of a unified diffusion framework.

39th Conference on Neural Information Processing Systems (NeurIPS 2025).

In this work, we present Riemannian Diffusion Language Model (RDLM), a continuous diffusion framework for language modeling that incorporates the geometry of the statistical manifold into the diffusion processes. We establish a connection between continuous flow on the statistical manifold and the discrete diffusion process, showing that the transition distribution can be modeled as a conditional flow on the manifold. Based on the analogy, we introduce a simple design of the diffusion processes on the manifold that generalizes previous discrete diffusion models. We further present a simulation-free training scheme that leverages radial symmetry, consisting of a simple parameterization and maximum likelihood-based training objectives. Through experiments on language modeling, image modeling, and biological sequence design, we validate that our framework outperforms existing discrete and continuous diffusion models.

## 2 Background

### 2.1 Discrete diffusion models

Discrete diffusion models [2, 39, 49, 52] define the diffusion process directly on discrete states using the Markov chains. The forward process describes the transition from the current state to other states, which is formalized by multiplying the transition matrix $Q_t$:

$$q(x_t|x_{t-1}) = \text{Cat}(x_t; Q_t x_{t-1}), \tag{1}$$

where $x_t$ is the random variable for the discrete states and $\text{Cat}(\cdot)$ denotes the categorical distribution. The marginal distribution corresponds to repeatedly multiplying transition matrices over time steps:

$$q(x_t|x) = \text{Cat}(x_t; \bar{Q}_t x) = \text{Cat}(x_t; Q_t \cdots Q_1 x). \tag{2}$$

Austin et al. [2] introduced several designs of the transition matrices, including masked (absorbing state) and uniform diffusion, and has been extended to continuous-time Markov chains (CTMC) [2, 6].

### 2.2 Statistical manifold of categorical distribution

Let $\mathcal{X} = \{1, \cdots, d\}$ denote the discrete data space, and let $\Delta^{d-1} = \{(p_1, \cdots, p_d) \in \mathbb{R}^d | \sum_i p_i = 1, p_i \geq 0\}$ denote the $(d-1)$-dimensional probability simplex. A categorical distribution over $\mathcal{X}$ can be parameterized by the parameters $p_1, \cdots, p_d$ satisfying $\sum_i p_i = 1$ and $p_i \geq 0$. The statistical manifold $\mathcal{P}(\mathcal{X})$ of the categorical distributions thus corresponds to the simplex $\Delta^{d-1}$ equipped with the Fisher-Rao metric [1, 47] (see Appendix A.1). There exists a diffeomorphism from the statistical manifold $\mathcal{P}(\mathcal{X})$ to the positive orthant of the $(d-1)$-dimensional sphere $\mathbb{S}_+^{d-1}$:

$$\pi : \mathcal{P}(\mathcal{X}) \to \mathbb{S}_+^{d-1}; \; p_i \mapsto u_i = \sqrt{p_i}, \tag{3}$$

which induces the geodesic distance $d_g(\boldsymbol{u}, \boldsymbol{v}) = \cos^{-1}\langle \boldsymbol{u}, \boldsymbol{v} \rangle$ for $\boldsymbol{u}, \boldsymbol{v} \in \mathbb{S}_+^{d-1}$, where $\langle \cdot, \cdot \rangle$ denotes the Euclidean inner product. We provide a more detailed explanation in Appendix A.1.

## 3 Riemannian Diffusion Language Model

We introduce a novel continuous diffusion model for language modeling. In this section, we present a single token generation framework, which we generalize to modeling sequences in Section 4.

### 3.1 Generalization of discrete diffusion

**Continuous reparameterization of discrete data**  To incorporate the geometry of the underlying categorical distribution, we leverage the statistical manifold to parameterize discrete data [12, 15]. Each point on the statistical manifold $\mathcal{P}(\mathcal{X})$ corresponds to the parameters of a categorical distribution over the discrete sample space $\mathcal{X} = \{1, \cdots, d\}$. In this way, discrete data can be represented as continuous parameters of categorical distributions on the manifold.

Yet the Fisher-Rao metric is ill-defined on the boundary of the manifold where the initial distribution of the parameterized data lies, leading to numerical instabilities near the boundary. To address this, we leverage the diffeomorphism $\pi$ (Eq. (3)) which maps $\mathcal{P}(\mathcal{X})$ to the positive orthant of a hypersphere $\mathbb{S}_+^{d-1}$ [12, 15], where each point $\boldsymbol{u} \in \mathbb{S}_+^{d-1}$ corresponds to $\text{Cat}(\cdot; \pi^{-1}(\boldsymbol{u}))$. This mapping enables

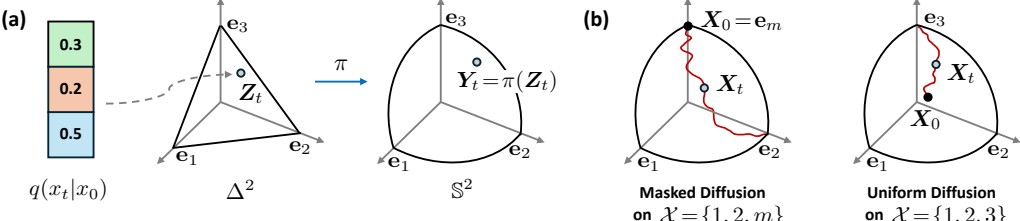

Figure 1: Illustration of the continuous reparameterization of discrete data and two types of our generative process on hypersphere. (a) Example of a transition distribution of a discrete diffusion process modeled by a continuous flow on a $d$-dimensional sphere. (b) Illustration of the diffusion processes on $\mathbb{S}^2$ generalizing masked diffusion and uniform diffusion, respectively.

discrete data to be reparameterized as continuous states on $\mathbb{S}^{d-1}$ while preserving the geometry of the categorical distribution, which we illustrate in Figure 1 (a). The reparameterized data distribution $p_{data}$ on the hypersphere can be written as $p_{data}(x) = \sum_{k=1}^{d} p_k \delta(x - e_k)$ where $p_k$ denotes the probability of the $k$-th state, and $e_k$ are $d$-dimensional one-hot vectors. In the case of masked diffusion, the discrete sample space is augmented with an additional mask state $m$.

**From discrete diffusion to continuous flow** Our key observation is that the transition distribution $q_t(x_t|x)$ of a discrete diffusion process (Eq. (2)) is a categorical distribution on $\mathcal{X}$. Therefore, modeling $q_t$ is equivalent to modeling continuous flow on the statistical manifold $\mathcal{P}(\mathcal{X})$. We show in the following proposition that discrete diffusion models over $\mathcal{X}$ can be modeled by a continuous flow on $\mathcal{P}(\mathcal{X})$ and further on $\mathbb{S}_+^{d-1}$ (we provide the full proof in Appendix A.2).

**Proposition 3.1.** *The transition distribution of discrete diffusion processes can be modeled by the continuous flow on the statistical manifold, and further on the hypersphere.*

*proof sketch.* A flow on $\mathbb{S}_+^{d-1}$ that interpolates $y_0$ and $y_1$ as geodesic is described by the ODE:

$$\frac{\mathrm{d}Y_t}{\mathrm{d}t} = -\frac{\mathrm{d}\log\kappa_t}{\mathrm{d}t}\exp_{Y_t}^{\text{-1}}(y_1), \quad Y_0 = y_0, \tag{4}$$

where $\exp^{\text{-1}}$ denotes the logarithm map on the hypersphere. Then, for a well-designed schedule $\kappa_t$ and endpoint $y_1$, the process $Z_t := \pi(Y_t)$ on $\mathcal{P}(\mathcal{X})$ corresponds to the transition distribution of the discrete diffusion process. In particular, we obtain the masked diffusion process for $y_1 = e_m$, i.e., the mask token, and the uniform diffusion process for $y_1 = \sum_{i=1}^{d} e_i/\sqrt{d}$. $\square$

Although discrete diffusion processes can be represented as a flow on the statistical manifold, this flow cannot be learned by a neural network. The network fails to generalize to points outside the geodesic that interpolates the prior and the data distribution, producing an incorrect vector field. Therefore, we present a simple design for the continuous diffusion model that generalizes existing discrete diffusion models.

### 3.2 Generative process on hypersphere

The task of modeling the distribution of discrete data can be reformulated as modeling a distribution $p_{data}$ on the hypersphere. Building upon the Riemannian diffusion mixture framework [33], we construct a diffusion process on $\mathbb{S}^{d-1}$ such that its terminal distribution matches $p_{data}$. The construction entails deriving a diffusion mixture representation based on bridge processes defined on $\mathbb{S}^{d-1}$.

We first derive a bridge process $\{\bar{X}_t\}_{t=0}^{T}$ on $\mathbb{S}^{d-1}$ from an arbitrary point $x_0 \in \mathbb{S}^{d-1}$ to $e_k$ as follows (we provide detailed derivation in Appendix A.3):

$$\mathrm{d}\bar{X}_t = \gamma_t \frac{\cos^{\text{-1}}\langle\bar{X}_t, e_k\rangle(e_k - \langle\bar{X}_t, e_k\rangle\bar{X}_t)}{\sqrt{1 - \langle\bar{X}_t, e_k\rangle^2}}\mathrm{d}t + \sigma_t\mathrm{d}\mathbf{B}_t^d, \quad \bar{X}_0 = x_0, \tag{5}$$

where $\gamma_t := \sigma_t^2 / \int_t^T \sigma_s^2 \mathrm{d}s$ and $\mathbf{B}_t^d$ denotes the Brownian motion defined on $\mathbb{S}^{d-1}$. Intuitively, the current state $X_t$ moves in the direction that minimizes the geodesic distance to the endpoint, resulting in a process that bridges the starting and end points. While different forms of the bridge process

exist, for example, scaling the drift or the diffusion coefficients, Eq. (5) yields a specific transition distribution that enables simulation-free training, which we explain in Section 3.3.

From the bridge processes, we construct a generative process $\{\boldsymbol{X}_t\}_{t=0}^{T}$ on $\mathbb{S}^{d-1}$ using the diffusion mixture representation (see Appendix A.4 for the formal definition of the diffusion mixture representation and the derivation of the generative process in Corollary A.8):

$$\mathrm{d}\boldsymbol{X}_t = \left[\sum_{k=1}^{d} p_{T|t}(\boldsymbol{e}_k|\boldsymbol{X}_t)\,\eta^k(\boldsymbol{X}_t,t)\right]\mathrm{d}t + \sigma_t\mathrm{d}\mathbf{B}_t^d,\ \ \boldsymbol{X}_0 = \boldsymbol{x}_0, \tag{6}$$

where $\eta^k(\cdot,t)$ denote the drift of the bridge process in Eq. (5). Here, $p_{T|t}(\boldsymbol{e}_k|\boldsymbol{X}_t)$ represents the probability that $\boldsymbol{e}_k$ will be the final outcome of the process at time $T$, given the current state $\boldsymbol{X}_t$ at time $t$. Note that the construction guarantees the terminal distribution of the process to be $p_{data}$.

An ideal generative process is one that gradually refines the uninformative states to recover the original tokens. We analyze the convergence of the bridge process through its radial process $r_t^k := d_g(\bar{\boldsymbol{X}}_t, \boldsymbol{e}_k)$ described by the following SDE (see Appendix A.3 for the derivation using Itô's formula):

$$\mathrm{d}r_t^k = \left[-\gamma_t r_t^k + \frac{(d-1)\sigma_t^2}{2}\cot r_t^k\right]\mathrm{d}t + \sigma_t\mathrm{d}W_t,\ \ r_0^k = \cos^{-1}\langle\boldsymbol{x}_0,\boldsymbol{e}_k\rangle, \tag{7}$$

where $W_t$ is a 1-dimensional Wiener process. For $\sigma_0 > \sigma_T$, the radial process converges rapidly in early time steps, making it difficult for a neural network to approximate accurately. We empirically find that the geometric schedule $\sigma_t = \sigma_0^{T-t}\sigma_T^t$ with $\sigma_0 < \sigma_T$ leads to gradual convergence.

**Masked diffusion** Based on Proposition 3.1, initializing the generative process in Eq. (6) with the mask token, i.e., $\boldsymbol{X}_0 = \boldsymbol{e}_m$, yields a mixture process that generalizes the discrete masked diffusion framework. The diffusion process starts at the mask token and progressively evolves toward one of the target tokens $\boldsymbol{e}_k$, as visualized in Figure 1 (b). From the perspective of the discrete diffusion model, our mixture process smoothly interpolates the discrete jump from $\boldsymbol{e}_m$ to $\boldsymbol{e}_k$ through intermediate continuous states $\boldsymbol{X}_t$, where the final token is determined by the probability $p_{T|t}(\boldsymbol{e}_k|\boldsymbol{X}_t)$.

The fundamental difference is that discrete masked diffusion operates through direct jumps between a token and the mask token, where any incorrect transition is irreversible. In contrast, our continuous approach allows for gradual transitions, providing numerous opportunities to correct wrong predictions during the process. This leads to more accurate modeling of the underlying data distribution.

**Uniform diffusion** Based on Proposition 3.1, the generalization of uniform diffusion can be achieved by initializing the generative process of Eq. (6) with the barycenter of the simplex $\Delta^{d-1}$ projected onto $\mathbb{S}^{d-1}$, i.e., $\boldsymbol{X}_0 = \pi(\sum_{i=1}^{d}\boldsymbol{e}_i/d) = \sum_{i=1}^{d}\boldsymbol{e}_i/\sqrt{d}$. We visualize the diffusion process in Figure 1 (b). Intuitively, the barycenter of $\Delta^{d-1}$ corresponds to the uniform categorical distribution over $d$ categories, which serves as the stationary distribution of the discrete uniform diffusion process.

**Mixture paths** We derive a new family of generative processes by constructing a mixture over the time marginals of generative processes $\{\mathbb{Q}_t^i : 1 \le i \le n\}$ (see Appendix A.5 for derivation):

$$\mathbb{Q}_t^{mix} := \sum_{i=1}^{n}\lambda_t^i\mathbb{Q}_t^i\ ,\ \ \sum_{i=1}^{n}\lambda_t^i = 1,\ 0 \le \lambda_t^i \le 1, \tag{8}$$

where $\lambda_t^i$ is the time-dependent mixing schedule assigned to the $i$-the generative path. This construction allows the resulting process to transition between different generative behaviors over time.

In particular, we propose a simple yet effective mixture path built from mixing the time marginals of the masked diffusion and uniform diffusion, for a time-dependent schedule $\lambda_t$ as follows:

$$\lambda_t\mathbb{Q}_t^{mask} + (1 - \lambda_t)\mathbb{Q}_t^{unif}, \tag{9}$$

with initial distribution $\lambda_0\delta(\boldsymbol{e}_m) + (1-\lambda_0)\delta(\sum_{i=1}^{d}\boldsymbol{e}_i/\sqrt{d})$. This formulation generalizes the mixture paths used in discrete flow matching [51] and the state-dependent schedule [52].

**Generalizing flow matching** Notably, our framework generalizes previous flow matching methods on the statistical manifold [12, 15]. By designing the noise schedule in Eq. (5) to be $\sigma_t \equiv \sigma_0 \to 0$, we obtain the conditional vector field of the flow matching models.

### 3.3 Simulation-Free Training with Radial Symmetry

Next, we introduce our training scheme. We present a simple parameterization of our generative model and derive the likelihood bound and training objectives. Further, we present a simulation-free training method based on the radial symmetry of the hypersphere.

**Model parameterization**    To use the diffusion process in Eq. (6) as a generative model, its unknown drift should be learned through a neural network, similarly to flow matching [8, 37] or bridge matching [33]. Yet the drift of the mixture process diverges near the terminal time $T$, which makes it challenging to learn. Therefore, instead of approximating the drift function directly, we propose to model the probability $p_{T|t}(\boldsymbol{X}_T|\boldsymbol{X}_t)$ with a neural network $\boldsymbol{s}_\theta$ as follows:

$$p_\theta(\boldsymbol{X}_t, t) \coloneqq \mathtt{softmax}\left(\boldsymbol{s}_\theta(\boldsymbol{X}_t, t)\right) = \left[p_{T|t}(\boldsymbol{e}_1|\boldsymbol{X}_t), \cdots, p_{T|t}(\boldsymbol{e}_d|\boldsymbol{X}_t)\right]^\mathrm{T}, \tag{10}$$

which converges to $\boldsymbol{e}_k$ for some $k$ as $t \to T$. In the case of masked diffusion, we set the probability $p_{T|t}(\boldsymbol{e}_m|\boldsymbol{X}_t)$ to be zero for all $t$, indicating that the final state cannot be a mask token. From Eq. (10), the drift of the mixture process in Eq. (6) is parameterized as follows:

$$\eta_\theta(\boldsymbol{X}_t, t) = \sum_{k=1}^{d} \left\langle p_\theta(\boldsymbol{X}_t, t), \boldsymbol{e}_k \right\rangle \eta^k(\boldsymbol{X}_t, t). \tag{11}$$

Our parameterization shares similar properties with the discrete masked diffusion [49]: (1) *Zero Mask Probabilities.* The final state cannot be a mask token. (2) *Carry-Over Unmasking.* If $\boldsymbol{X}_t$ converges to a token $\boldsymbol{e}_k$ before the terminal time, $\eta_\theta$ converges to zero, and the state $\boldsymbol{X}_t$ is carried over without changing to different token.

**Likelihood bound**    We derive a tractable upper bound on the negative likelihood of our generative model by applying the Girsanov theorem on compact manifolds (De Bortoli et al. [16], Corollary H.3). Specifically, we first establish a point-wise upper bound on the negative log-likelihood under the parameterized mixture process $\mathbb{Q}^\theta$, using the KL divergence between $\mathbb{Q}^\theta$ and a bridge process $\mathbb{Q}^k$, which is conditioned on endpoints $\boldsymbol{x}_0$ and $\boldsymbol{e}_k$. Applying the Girsanov theorem, we obtain the following variational upper bound (we provide a detailed derivation in Appendix A.6):

$$-\log \hat{p}_\theta(\boldsymbol{e}_k) = D_{KL}(\mathbb{Q}_T^k \| \mathbb{Q}_T^\theta) \leq \mathbb{E}_{\boldsymbol{X} \sim \mathbb{Q}^k} \left[ \frac{1}{2} \int_0^T \left\| \sigma_t^{-1} \Big( \eta_\theta(\boldsymbol{X}_t, t) - \eta^k(\boldsymbol{X}_t, t) \Big) \right\|_2^2 \mathrm{d}t \right] \tag{12}$$

where $\eta^k$ is the drift defined in Eq. (5). The point-wise likelihood bound yields an upper bound on the negative log-likelihood of our generative model parameterized by $p_\theta$:

$$\mathbb{E}_{\boldsymbol{z} \sim p_{data}}\big[ -\log \hat{p}_\theta(\boldsymbol{z}) \big] \leq \mathbb{E}_{\substack{\boldsymbol{e}_k \sim p_{data} \\ \boldsymbol{X} \sim \mathbb{Q}^k}} \left[ \frac{1}{2} \int_0^T \left\| \sigma_t^{-1} \Big( \eta_\theta(\boldsymbol{X}_t, t) - \eta^k(\boldsymbol{X}_t, t) \Big) \right\|_2^2 \mathrm{d}t \right]. \tag{13}$$

**Objective**    Based on the likelihood bound in Eq. (13), we introduce a maximum likelihood training objective for the model parameterization $p_\theta$ in Eq. (10):

$$\mathcal{L}(\theta) = \mathbb{E}_{\substack{\boldsymbol{e}_k \sim p_{data} \\ \boldsymbol{X} \sim \mathbb{Q}^k}} \left[ \frac{1}{2} \int_0^T \sigma_t^{-2} \left\| \sum_{l=1}^{d} \left\langle p_\theta(\boldsymbol{X}_t, t), \boldsymbol{e}_l \right\rangle \eta^l(\boldsymbol{X}_t, t) - \eta^k(\boldsymbol{X}_t, t) \right\|_2^2 \mathrm{d}t \right]. \tag{14}$$

This objective corresponds to minimizing the mean squared error in approximating the drift term.

In particular, $\mathcal{L}(\theta)$ can be minimized by reducing the cross-entropy between the predicted probability $p_\theta(\boldsymbol{X}_t, t)$ and the target one-hot vector $\boldsymbol{e}_k$. Therefore we present a cross-entropy-based training objective, analogous to those used in discrete diffusion models [49, 52]:

$$\mathcal{L}^{CE}(\theta) = \mathbb{E}_{\substack{\boldsymbol{e}_k \sim p_{data} \\ \boldsymbol{X} \sim \mathbb{Q}^k}} \left[ \int_0^T -\log \left\langle p_\theta(\boldsymbol{X}_t, t), \boldsymbol{e}_k \right\rangle \mathrm{d}t \right]. \tag{15}$$

We show in Appendix A.7 that minimizing the cross-entropy-based objective in Eq. (15) leads to minimizing $\mathcal{L}(\theta)$, thereby ensuring maximum likelihood training. We experimentally find that the cross-entropy loss $\mathcal{L}^{CE}(\theta)$ yields faster convergence in training and leads to better performance than the mean squared error loss $\mathcal{L}(\theta)$.

**Importance sampling** The difficulty of approximating the probability $p_{T|t}(\boldsymbol{X}_T|\boldsymbol{X}_t)$ varies significantly across different time points $t$. While predicting $\boldsymbol{X}_T$ is fairly easy in the later stage of the process, it is challenging to do so during the middle of the process. The training can be improved by training more on the challenging time points. We derive an equivalent objective by applying importance sampling over $t$, which reweights the time distribution to focus on a specific interval:

$$\mathcal{L}_q^{CE}(\theta) = \mathbb{E}_{\substack{\boldsymbol{e}_k \sim p_{data} \\ \boldsymbol{X} \sim \mathbb{Q}^k}} \mathbb{E}_{t \sim q} \Big[ - q(t)^{-1} \log \big\langle p_\theta(\boldsymbol{X}_t, t), \boldsymbol{e}_k \big\rangle \Big] \tag{16}$$

where $q$ is a normalized proposal distribution over $t$. We find that a simple choice $q(t) = \epsilon + (1 - 2\epsilon)\mathbf{1}_{[a,b]}(t)$ with small $\epsilon$ effectively concentrates sampling within the desired time interval.

**Approximation of transition distribution** Our training objective involves sampling $\boldsymbol{X}_t$ from the bridge processes at each iteration. Yet this introduces a significant bottleneck during training, as it requires simulating the process due to its intractable transition distribution on the $d$-dimensional sphere. Therefore, we present an approximate sampling method that bypasses the need for simulation, thereby enabling scalable training across large vocabularies.

We propose to approximate the distribution $p(\boldsymbol{X}_t|\boldsymbol{X}_0, \boldsymbol{X}_T)$ as the push-forward of a Gaussian distribution on the tangent space via the exponential map, i.e., the Riemannian normal. This approximation is justified by the fact that Eq. (5) results from applying a time change [64] to a simple bridge process (Eq. (59)), which yields a transition distribution similar to Riemannian normal.

We parameterize the mean of the Riemannian normal distribution as $\boldsymbol{\mu}_t \coloneqq \mathbb{E}\boldsymbol{X}_t / \|\mathbb{E}\boldsymbol{X}_t\|$ and its covariance $\boldsymbol{\Sigma}_t \coloneqq \mathrm{Cov}\big[\exp_{\boldsymbol{\mu}_t}^{-1}(\boldsymbol{X}_t)\big]$, using the parameters $\alpha_t$ and $\rho_t$ as follows:

$$\boldsymbol{\mu}_t = \frac{\alpha_t}{\sin \phi_0} \boldsymbol{X}_T + \left( \sqrt{1 - \alpha_t^2} - \frac{\alpha_t \cos \phi_0}{\sin \phi_0} \right) \boldsymbol{X}_0 \,, \quad \boldsymbol{\Sigma}_t = \rho_t^2 \mathbf{I}, \tag{17}$$

where $\phi_0 \coloneqq \cos^{-1}\langle \boldsymbol{X}_0, \boldsymbol{X}_T \rangle$. Intuitively, $\boldsymbol{\mu}_t$ represents the normalized centroid of the samples $\boldsymbol{X}_t$, and $\boldsymbol{\Sigma}_t$ captures to the covariance of the lifted samples in the tangent space $\mathcal{T}_{\boldsymbol{\mu}_t}$.

**Parameters of Riemannian normal** While the parameters $\alpha_t$ and $\rho_t$ are generally intractable, we derive them from the 1-dimensional projections of the mixture process. Our main idea is to express the parameters in terms of the projected processes $z_t^T \coloneqq \langle \boldsymbol{X}_{t|0,T}, \boldsymbol{X}_T \rangle$ and $z_t^0 \coloneqq \langle \boldsymbol{X}_{t|0,T}, \boldsymbol{X}_0 \rangle$, where $\boldsymbol{X}_{t|0,T}$ denotes the diffusion process $\{\boldsymbol{X}_t\}_{t=0}^T$ conditioned on fixed endpoints $\boldsymbol{X}_0$ and $\boldsymbol{X}_T$. These projected processes are modeled by the following 1-dimensional SDEs (see Appendix A.8 for the derivation using the Itô's formula and the radial symmetry of $\mathbb{S}^{d-1}$):

$$\mathrm{d}z_t^T = \left[ \gamma_t \cos^{-1} z_t^T \sqrt{1 - (z_t^T)^2} - \frac{(d-1)\sigma_t^2}{2} z_t^T \right] \mathrm{d}t + \sigma_t \sqrt{1 - (z_t^T)^2} \, \mathrm{d}W_t^T, \tag{18}$$

$$\mathrm{d}z_t^0 = \left[ \gamma_t \frac{\cos^{-1} z_t^T}{\sqrt{1 - (z_t^T)^2}} \left( z_0^T - z_t^0 z_t^T \right) - \frac{(d-1)\sigma_t^2}{2} z_t^0 \right] \mathrm{d}t + \sigma_t \sqrt{1 - (z_t^0)^2} \, \mathrm{d}W_t^0, \tag{19}$$

with $z_0^T = \langle \boldsymbol{X}_0, \boldsymbol{X}_T \rangle$ and $z_0^0 = 1$, where $W_t^T$ and $W_t^0$ denote 1-dimensional Wiener processes. In the case of masked and uniform diffusion, $\boldsymbol{X}_0$ is fixed to a single point such that $\langle \boldsymbol{X}_0, \boldsymbol{e}_k \rangle$ is identical for all non-mask tokens $\boldsymbol{e}_k$. As a result, the mean projections $\mathbb{E}z_t^T$ and $\mathbb{E}z_t^0$ remain invariant with respect to the choice of $\boldsymbol{X}_T$.

Based on the radial symmetry of $\mathbb{S}^{d-1}$, we derive the parameters $\alpha_t$ and $\rho_t$ from the mean projections $\mathbb{E}z_t^0$ and $\mathbb{E}z_t^T$ as follows (we provide detailed derviation in Appendix A.9):

$$\alpha_t = \sqrt{\frac{(\mathbb{E}z_t^T / \mathbb{E}z_t^0 - \cos \phi_0)^2}{\sin^2 \phi_0 + (\mathbb{E}z_t^T / \mathbb{E}z_t^0 - \cos \phi_0)^2}} \,, \quad \rho_t = F_d^{-1}\left( \mathbb{E}z_t^0 / \sqrt{1 - \alpha_t^2} \right), \tag{20}$$

where $\phi_0 \coloneqq \cos^{-1}\langle \boldsymbol{X}_0, \boldsymbol{X}_T \rangle$ and $F_d^{-1}$ denotes the inverse of a damped Kummer function (Eq. (115)). For small values of $d$, we calibrate $\rho_t$ by applying a constant scaling factor.

The mean projections $\mathbb{E}z_t^0$ and $\mathbb{E}z_t^T$ can be easily obtained by simulating the 1-dimensional processes Eq. (18) and Eq. (19). Therefore, prior to training our model $p_\theta$, we precompute the parameters $\{\alpha_{i/K}, \rho_{i/K}\}_{i=0}^K$ once, using a sufficiently large value of $K$. The procedure for this precomputation is outlined in Algorithm 3 in the Appendix.

---

**Algorithm 1** Training

---

**Input:** Initial point $\boldsymbol{u}$, model $p_\theta$, vocabulary size $d$, token sequence length $L$, time distribution $q(t)$, pre-computed $\{\alpha_{i/K}, \rho_{i/K}\}_{i=0}^K$

**For each epoch:**

1: Sample token sequence $\boldsymbol{s}$ from the training set
2: $\boldsymbol{X}_0 \leftarrow (\boldsymbol{u})^L$ and $\boldsymbol{X}_1 \leftarrow \left(\text{ONE-HOT}(\boldsymbol{s}^i, d)\right)_{i=1}^L$
3: $\phi_0 \leftarrow \left(\cos^{-1}\left\langle \boldsymbol{X}_0^i, \boldsymbol{X}_1^i \right\rangle\right)_{i=1}^L$
4: $t \sim q$ and $\alpha_t, \rho_t \leftarrow \text{INTERPOLATE}\left(\{\alpha_{i/K}, \rho_{i/K}\}_{i=0}^K\right)$
5: $\boldsymbol{\mu}_t \leftarrow \left(\frac{\alpha_t}{\sin\phi_0^i}\boldsymbol{X}_1^i + \left(\sqrt{1-\alpha_t^2} - \frac{\alpha_t \cos\phi_0^i}{\sin\phi_0^i}\right)\boldsymbol{X}_0^i\right)_{i=1}^L$    $\triangleright$ Eq. (17)
6: $\boldsymbol{X}_t \sim \mathcal{N}_{\mathbb{S}^{d-1}}(\boldsymbol{\mu}_t^1, \rho_t^2 \mathbf{I}_d) \times \cdots \times \mathcal{N}_{\mathbb{S}^{d-1}}(\boldsymbol{\mu}_t^L, \rho_t^2 \mathbf{I}_d)$    $\triangleright$ Sample from Riemannian normal
7: $\mathcal{L}_\theta \leftarrow -q(t)^{-1}\log\left\langle p_\theta(\boldsymbol{X}_t, t), \boldsymbol{X}_1\right\rangle$    $\triangleright$ Cross-entropy-based loss in Eq. (16)
8: Update $\theta$ using $\mathcal{L}_\theta$

---

**Algorithm 2** Sampling

---

**Input:** Initial point $\boldsymbol{u}$, trained model $p_\theta$, vocabulary size $d$, number of sampling steps $M$, token sequence length $L$, noise schedule $\sigma_t$

1: $\boldsymbol{X} \sim (\boldsymbol{u})^L$, $t \leftarrow 0$ and $\delta t \leftarrow 1/M$    $\triangleright$ Start from the initial point
2: **for** $m = 1$ **to** $M$ **do**
3:    $\mathbf{w} \sim \left(\mathcal{N}(0, \mathbf{I}_d)\right)^L$
4:    $p \leftarrow p_\theta(\boldsymbol{X}, t)$
5:    $\eta_\theta \leftarrow \left(\sum_{k=1}^d \left\langle p^i, \boldsymbol{e}_k\right\rangle \gamma_t \frac{\cos^{-1}\langle \boldsymbol{X}^i, \boldsymbol{e}_k\rangle (\boldsymbol{e}_k - \langle \boldsymbol{X}^i, \boldsymbol{e}_k\rangle \boldsymbol{X}^i)}{\sqrt{1-\langle \boldsymbol{X}^i, \boldsymbol{e}_k\rangle^2}}\right)_{i=1}^L$    $\triangleright$ Parameterization in Eq. (11)
6:    $\boldsymbol{X} \leftarrow \left(\exp_{\boldsymbol{X}^i}\left(\eta_\theta^i \delta t + \sigma_t\sqrt{\delta t}\mathbf{w}^i\right)\right)_{i=1}^L$    $\triangleright$ Geodesic random walk
7:    $t \leftarrow t + \delta t$
8: **end for**
9: $\boldsymbol{s} \leftarrow \left(\text{ARGMAX}(\boldsymbol{X}^i)\right)_{i=1}^L$
10: **Return:** Token sequence $\boldsymbol{s}$

---

During training, we can sample $\boldsymbol{X}_t$ from the Riemannian normal distribution without expensive simulation of the bridge processes. Compared to the simulation-based training, our approach yields a $\times 50$ speedup. In Section 6.4, we experimentally demonstrate that the Riemannian normal provides an accurate approximation of the distribution of $\boldsymbol{X}_t$.

## 4   Generation of Token Sequences

**Modeling sequence of tokens**   We now extend the single-token modeling framework to the generation of token sequences. Since each token in the sequence is reparameterized onto a $d$-dimensional sphere, a sequence of length $n$ is modeled on the product manifold $(\mathbb{S}^{d-1})^n$. This formulation allows the sequence-level diffusion to be treated as a joint process over the spherical components.

We model the generative process as a system of $n$ SDEs $\{(\boldsymbol{X}_t^1, \cdots, \boldsymbol{X}_t^n)\}_{t=0}^T$, where each $\boldsymbol{X}_t^i$ evolves according to a diffusion process on $\mathbb{S}^{d-1}$, analogous to the single-token formulation in Eq. (6):

$$\mathrm{d}\boldsymbol{X}_t^i = \left[\sum_{k=1}^d p(\boldsymbol{X}_T^i = \boldsymbol{e}_k | \boldsymbol{X}_t^{1:n})\, \eta^k(\boldsymbol{X}_t^i, t)\right]\mathrm{d}t + \sigma_t \mathrm{d}\mathbf{B}_t^d,\ 1 \leq i \leq n. \tag{21}$$

Here $p(\boldsymbol{X}_T^i = \boldsymbol{e}_k | \boldsymbol{X}_t^{1:n})$ denotes the probability that the $i$-th token corresponds to the $k$-th state, conditioned on the current intermediate sequence $\boldsymbol{X}_t^{1:n}$. Using the parameterization defined in Eq. (10), we train a neural network to predict $p(\boldsymbol{X}_T^{1:n} | \boldsymbol{X}_t^{1:n})$. The training and sampling procedures for modeling token sequences are outlined in Algorithms 1 and 2, respectively.

**Dimension splitting of statistical manifold**    For a large vocabulary set, the corresponding statistical manifold becomes high-dimensional, which introduces two challenges: (1) *Sharp transition.* Bridge processes on high-dimensional spheres tend to exhibit sharp transitions near the terminal time. This high-dimensional convergence behavior makes the mixture process difficult for neural networks to learn. (2) *High input dimensionality.* The input to the network resides in a high-dimensional space, requiring sufficiently large hidden dimensions to encode the data adequately. Models with limited capacity fail to learn the conditional probability $p(\boldsymbol{X}_T^{1:n}|\boldsymbol{X}_t^{1:n})$.

To address these challenges, we introduce *dimension splitting*, a simple technique to reduce the dimensionality of the parameterized manifold. Instead of mapping the $k$-th token directly to $\mathbb{S}^{d-1}$, we first represent the index $k$ in base $b$, and then map the represntation to the product manifold $(\mathbb{S}^b)^m$ for $m := \lceil \log_b d \rceil$. Dimension splitting reparameterizes a sequence of length $L$ to a product manifold $(\mathbb{S}^b)^{mL}$. The resulting bridge processes on $\mathbb{S}^b$ with small $b$ exhibit gradual convergence over time, making them significantly easier for neural networks to learn. Dimension splitting significantly enhances the likelihood when used together with the mixture path (Eq. (9)).

## 5    Related Work

**Discrete diffusion models**    Discrete diffusion directly models the Markov chain on the discrete data space. One-hot data distributions are gradually corrupted to a stationary distribution using specific transition matrices, and the noising process corresponds to the stochastic jumps between states in the Markov chain. D3PM [2] introduces discrete-time Markov forward processes with both uniform and absorbing state transition matrices, and has been generalized to the continuous-time Markov chain framework [6]. SEDD [39] proposes learning the score entropy of discrete states instead of predicting the mean. Recent works [49, 52] introduce continuous-time masked diffusion models, which offer simpler likelihood bounds compared to previous works. We provide further discussions on comparison with discrete diffusion models in Appendix A.10.

**Continuous diffusion models for discrete data**    Early approaches to discrete data modeling either fully relaxed discrete data into continuous space [24] or embedded tokens into a latent space [18, 36], without imposing any constraint. However, continuous relaxation without constraint fails to capture the discreteness of the categorical distribution. Recent works operate directly in logit space [21, 29] or on the probability simplex [3, 54], but rely on imperfect assumptions that fail to accurately represent the underlying categorical distribution. Flow matching has been applied to the statistical manifold to model the categorical distribution [12, 15], but these methods are limited to short sequences and small vocabularies. We provide a detailed comparison in Appendix A.10.

## 6    Experiments

### 6.1    Text generation

We evaluate our Riemannian Diffusion Language Model (RDLM) for text generation tasks on two language benchmarks: Text8 [42] and One Billion Words Dataset [7].

**Baselines**    We compare against state-of-the-art diffusion and autoregressive models. Multinomial Diffusion [29], D3PM [2], SEDD [39], MDLM [49], MD4 [52] are discrete diffusion models. Plaid [23] and Bayesian Flow Network (BFN) [21] are continuous diffusion models. IAF/SCF [63], AR Argmax Flow [29], and Discrete Flow [58] are flow-based models, and ARDM [30] and MAC [53] are any-order autoregressive models. We also compare with the transformer AR model [61]. We provide further details on the baselines in Appendix B.2

**Implementation details**    For all experiments, we use the same data split and context size following Lou et al. [39] and Sahoo et al. [49]. For Text8, we randomly sample contiguous chunks of length 256 as done in previous works [2, 39]. For One Billion Words, we use the same tokenizer as in He et al. [25] with context size 128. We use a diffusion transformer architecture [44] with rotary positional embeddings [55] for all the experiments and match the number of parameters as used in the previous works [39, 49]. For our model, we use the mixture path of masked and uniform diffusion (Eq. (9)) and apply dimension splitting for a large vocabulary. We provide more details in Appendix B.2.

Table 2: **Test perplexity** results on LM1B dataset. Baseline results taken from Sahoo et al. [49].

| Method | # Param. | PPL ($\downarrow$) |
|---|---|---|
| *Autoregressive* | | |
| Transformer-X Base [14] | 0.46B | 23.5 |
| OmniNet$_T$ [57] | 100M | 21.5 |
| Transformer [61] | 110M | 22.32 |
| *Discrete Diffusion* | | |
| BERT-Mouth [62] | 110M | $\leq 142.89$ |
| D3PM Absorb [2] | 70M | $\leq 76.90$ |
| DiffusionBert [25] | 110M | $\leq 63.78$ |
| SEDD [39] | 110M | $\leq 32.79$ |
| MDLM [49] | 110M | $\leq 27.04$ |
| *Continuous Diffusion* | | |
| Diffusion-LM [36] | 80M | $\leq 118.62$ |
| RDLM (Ours) | 110M | $\leq 28.44$ |

Table 3: **BPD** results on CIFAR-10 test set. Baseline results taken from Shi et al. [52].

| Method | # Param. | BPD ($\downarrow$) |
|---|---|---|
| *Autoregressive* | | |
| PixelRNN [60] | | 3.00 |
| Gated PixelCNN [59] | | 3.03 |
| PixelCNN++ [50] | 53M | 2.92 |
| PixelSNAIL [11] | 46M | 2.85 |
| Image Transformer [43] | | 2.90 |
| Sparse Transformer [13] | 59M | 2.80 |
| *Discrete Diffusion* | | |
| D3PM Absorb [2] | 37M | $\leq 4.40$ |
| D3PM Gauss [2] | 36M | $\leq 3.44$ |
| $\tau$LDR [6] | 36M | $\leq 3.59$ |
| $\tau$LDR Absorb [6] | 36M | $\leq 3.52$ |
| MD4 [52] | 28M | $\leq 2.78$ |
| *Continuous Diffusion* | | |
| RDLM (Ours) | 28M | $\leq \mathbf{2.73}$ |

**Text8** We first evaluate on a small-scale character-level language modeling task. The Text8 [42] dataset is a character-level text modeling benchmark extracted from English Wikipedia. We train models on short text chunks of length 256 and evaluate the performance using Bits Per Character (BPC). As shown in Table 1, our framework outperforms all previous diffusion models, including both discrete and continuous methods. We also outperform any-order autoregressive models that generate texts in flexible decoding order, similar to discrete diffusion models. We achieve similar generative perplexity and entropy compared to existing discrete diffusion models. We provide generated texts from RDLM in Appendix C.1.

**One Billion Words** We further evaluate RDLM on One Billion Words Dataset (LM1B) [7], a medium-scale real-world language benchmark with a vocabulary size of 30522. We evaluate the performance using perplexity (PPL), and the results are summarized in Table 2. RDLM outperforms most existing diffusion models and is competitive with the state-of-the-art discrete diffusion model [49]. Notably, ours significantly outperforms the prior continuous diffusion model [36], demonstrating the effectiveness of incorporating the geometry of the underlying categorical distribution. We provide a discussion of the results with MDLM [49] in Appendix B.2. The generated texts are presented in Appendix C.2.

Table 1: **Bits Per Character (BPC)** results on Text8 test set. Results are taken from the corresponding papers.

| Method | BPC ($\downarrow$) |
|---|---|
| *Autoregressive* | |
| AR Argmax Flow [29] | 1.39 |
| Transformer AR [61] | **1.23** |
| Discrete Flow [58] | **1.23** |
| *Any-order Autoregressive* | |
| ARDM [30] | $\leq 1.43$ |
| MAC [53] | $\leq 1.40$ |
| *Discrete Diffusion* | |
| Multinomial Diffusion [29] | $\leq 1.72$ |
| D3PM Uniform [2] | $\leq 1.61$ |
| D3PM Absorb [2] | $\leq 1.45$ |
| SEDD Absorb [39] | $\leq 1.39$ |
| MDLM [49] | $\leq 1.40$ |
| MD4 [52] | $\leq 1.37$ |
| *Continuous Diffusion* | |
| Plaid [23] | $\leq 1.48$ |
| BFN [21] | $\leq 1.41$ |
| RDLM (Ours) | $\leq \mathbf{1.32}$ |

## 6.2 Pixel-level image modeling

We further explore applications of RDLM beyond the text domain by applying it to order-agnostic image data. Each image is represented as a set of discrete tokens with a vocabulary of size 256, removing information about pixel proximity. Note that this is different from the experimental settings with image diffusion models [27, 34] that use spatial information. We compare RDLM against autoregressive models and discrete diffusion models that operate directly on raw pixel space, which we describe in Appendix B.3. As shown in Table 3, our method achieves the lowest Bits Per Dimension (BPD), outperforming the discrete diffusion models [2, 52] and autoregressive baselines [11, 13]. We attribute this strong performance on inherently continuous data to the continuous nature of our framework, which fully exploits iterative refinement, suggesting its potential for unifying modeling across different modalities.

## 6.3 DNA sequence design

We demonstrate that our framework can be applied to biological sequence generation. We evaluate our method on the promoter DNA sequence design task, which aims to generate valid promoter DNA sequences conditioned on transcription profiles. A detailed description of the task is provided in Appendix B.4. Model performance is measured by the mean squared error (MSE) between the predicted regulatory activity of the generated sequence and that of the original sequence corresponding to the transcription profile. Table 4 shows that our framework achieves the lowest MSE, outperforming the flow matching methods [15, 54] and the discrete diffusion model [2].

Table 4: **MSE** results on the generated promoter DNA sequences. Baseline results are taken from Davis et al. [15].

| Method | MSE ($\downarrow$) |
| --- | --- |
| Bit-Diffusion (bit) [10] | 0.041 |
| Bit-Diffusion (one-hot) [10] | 0.040 |
| D3PM Uniform [2] | 0.038 |
| DDSM [3] | 0.033 |
| DirichletFM [54] | 0.034 |
| Language Model | 0.034 |
| Fisher-Flow [15] | 0.029 |
| RDLM (Ours) | **0.027** |

## 6.4 Analysis

**Training objective** We validate the effectiveness of our cross-entropy-based loss of Eq. (15) in Table 5. Compared to the mean squared error loss of Eq. (14), the cross-entropy loss provides faster convergence in training and better NLL. Furthermore, Table 5 shows that applying importance sampling to the training objective as defined in Eq. (16) yields improved likelihood.

**Approximation of transition distribution** We validate that our approximate sampling method closely matches the true transition distribution of the mixture process. In Figure 2, we report the maximum mean discrepancy (MMD) [22] distance between the simulated transition distribution and the approximated distribution obtained using the Riemannian normal. The approximated distributions exhibit nearly identical MMD as the simulated distributions, indicating that he approximation is accurate and reliable. Notably, the discrepancy approaches zero in high-dimensional manifolds, where simulation becomes increasingly expensive, making simulation-based training impractical.

**Dimension splitting** For datasets with a large vocabulary, such as the LM1B dataset, our dimension splitting technique (Section 4) results in a significant improvement. Table 6 shows that directly training a model on discrete data with a large vocabulary fails to capture the underlying distribution, due to the high input dimensionality. In particular, the sharp transition near the terminal time for a high-dimensional mixture process makes it challenging for neural networks to learn. In large vocabulary settings, we achieve the best result via dimension splitting, combined with modeling the generative process using a mixture path of masked and uniform diffusion.

## 7 Conclusion

In this work, we introduced the Riemannian Diffusion Language Model (RDLM), a continuous diffusion model for language and discrete data. We present a simple framework that generalizes discrete diffusion models, building on the connection between the transition distribution and continuous flow on the statistical manifold. We provide general designs for generative processes and introduce a simulation-free training scheme leveraging the radial symmetry. Through experiments on language modeling benchmarks, RDLM demonstrates strong performance over prior discrete and continuous diffusion models. We further extend our approach to other modalities, including image and biological sequence generation, where RLDM achieves consistently strong results.

## 8 Acknowledgements

This work was supported by National Research Foundation of Korea (NRF) grant funded by the Korea government (MSIT) (No. RS-2023-00256259), Institute for Information & communications Technology Promotion(IITP) grant funded by the Korea government(MSIT) (No.RS-2019-II190075 Artificial Intelligence Graduate School Program(KAIST)), Information & Communications Technology Planning & Evaluation (IITP) with a grant funded by the Ministry of Science and ICT (MSIT) of the Republic of Korea in connection with the Global AI Frontier Lab International Collaborative Research. (No. RS-2024-00469482 & RS-2024-00509279), and artificial intelligence industrial convergence

cluster development project funded by the Ministry of Science and ICT(MSIT, Korea)&Gwangju Metropolitan City.

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

# Appendix

## A  Derivations

### A.1  Preliminaries

**Statistical Manifold of Categorical Distributions**  For a discrete sample space $\mathcal{X} = \{1, 2, \cdots, d\}$, a $d$-class categorical distribution over $\mathcal{X}$ is parameterized by $d$ number of parameters $p_1, \cdots, p_d \geq 0$ such tat $\sum_{i=1}^{d} p_i = 1$. The parameter space corresponds to the $(d-1)$-dimensional probability simplex:

$$\Delta^{d-1} = \left\{ (p_1, \cdots, p_d) \in \mathbb{R}^d : \sum_{i=1}^{d} p_i = 1, p_i \geq 0 \right\}, \tag{22}$$

A natural choice of a Riemannian metric on the simplex is the Fisher-Rao metric [1, 47]. For an interior point $\boldsymbol{p} \in \Delta^{d-1}$, the Fisher-Rao metric is defined as follows:

$$g_{FR}(\boldsymbol{p})[\boldsymbol{x}, \boldsymbol{y}] := \langle \boldsymbol{x}, \boldsymbol{y} \rangle_{\boldsymbol{p}} := \left\langle \frac{\boldsymbol{x}}{\sqrt{\boldsymbol{p}}}, \frac{\boldsymbol{y}}{\sqrt{\boldsymbol{p}}} \right\rangle = \sum_{i=1}^{d} \frac{x_i y_i}{p_i}, \quad \boldsymbol{x}, \boldsymbol{y} \in \mathcal{T}_{\boldsymbol{p}} \Delta^{d-1}, \tag{23}$$

where the normalization by $\sqrt{\boldsymbol{p}}$ in the inner product is performed component-wise. This induces a geodesic distance on the simplex defined as follows:

$$d(\boldsymbol{p}, \boldsymbol{q}) = 2 \cos^{-1} \left( \sum_{i=1}^{d} \sqrt{p_i q_i} \right), \quad \boldsymbol{p}, \boldsymbol{q} \in \Delta^{d-1}, \tag{24}$$

where $\boldsymbol{p}$ and $\boldsymbol{q}$ corresponds to the parameters of categorical distributions. The probability simplex $\Delta^{d-1}$ equipped with the Fisher-Rao metric is a Riemannian manifold called the statistical manifold of categorical distribution, denoted as $\mathcal{P}(\mathcal{X})$ throughout the paper. The tangent space at an interior point $\boldsymbol{p}$ is identified as $\mathcal{T}_{\boldsymbol{p}}(\mathcal{P}(\mathcal{X})) = \left\{ \boldsymbol{x} \in \mathbb{R}^d : \sum_{i=1}^{d} \boldsymbol{x}_i = 0 \right\}$. For further details on the geometry of the statistical manifold, we refer the reader to Ay et al. [4].

**Hypersphere**  $\mathbb{S}^{d-1}$ denotes the $(d-1)$-dimensional sphere $\left\{ \boldsymbol{u} = (u_1, \cdots, u_d) : \sum_i u_i^2 = 1 \right\}$ and $\mathbb{S}_+^{d-1} = \left\{ \boldsymbol{u} = (u_1, \cdots, u_d) : \sum_i u_i^2 = 1, u_i \geq 0 \right\}$ denotes a positive orthant of $\mathbb{S}^{d-1}$. The hypersphere $\mathbb{S}^{d-1}$ can be embedded into the ambient Euclidean space $\mathbb{R}^d$, which induces a canonical inner product $\langle \boldsymbol{x}, \boldsymbol{y} \rangle := \sum_{i=1}^{d} x_i y_i$. For a discrete sample space $\mathcal{X} = \{1, 2, \cdots, d\}$, there exists a diffeomorphism from $\mathcal{P}(\mathcal{X})$ to $\mathbb{S}_+^{d-1}$ defined as follows:

$$\pi : \mathcal{P}(\mathcal{X}) \to \mathbb{S}_+^{d-1} \;\; ; \;\; p_i \mapsto u_i = \sqrt{p_i},$$
$$\pi^{-1} : \mathbb{S}_+^{d-1} \to \mathcal{P}(\mathcal{X}) \;\; ; \;\; u_i \mapsto p_i = u_i^2. \tag{25}$$

The diffeomorphism induces the the geodesic distance on $\mathbb{S}_+^{d-1}$:

$$d_g(\boldsymbol{u}, \boldsymbol{v}) = \cos^{-1}\langle \boldsymbol{u}, \boldsymbol{v} \rangle, \quad \boldsymbol{u}, \boldsymbol{v} \in \mathbb{S}_+^{d-1}, \tag{26}$$

for which the geodesic corresponds to the great circle connecting two points $\boldsymbol{u}$ and $\boldsymbol{v}$. The corresponding exponential and logarithm maps on $\mathbb{S}^{d-1}$ can be computed as follows:

$$\exp_{\boldsymbol{u}} \boldsymbol{x} = \cos(\|\boldsymbol{x}\|)\boldsymbol{u} + \sin(\|\boldsymbol{x}\|)\frac{\boldsymbol{x}}{\|\boldsymbol{x}\|}, \quad \boldsymbol{u} \in \mathbb{S}^{d-1}, \boldsymbol{x} \in \mathcal{T}_{\boldsymbol{u}}(\mathbb{S}^{d-1}), \tag{27}$$

$$\exp_{\boldsymbol{u}}^{-1}(\boldsymbol{v}) = \frac{\cos^{-1}\langle \boldsymbol{u}, \boldsymbol{v} \rangle}{\sqrt{1 - \langle \boldsymbol{u}, \boldsymbol{v} \rangle^2}} \Big( \boldsymbol{v} - \langle \boldsymbol{u}, \boldsymbol{v} \rangle \boldsymbol{u} \Big), \quad \boldsymbol{u}, \boldsymbol{v} \in \mathbb{S}^{d-1}. \tag{28}$$

Additionally, define the radial distance $r^{\boldsymbol{v}}(\boldsymbol{x}) := d_g(\boldsymbol{x}, \boldsymbol{v}) \in \mathbb{R}$ where $d_g$ denotes the geodesic distance on $\mathbb{S}^{d-1}$. Then we have the following identities:

$$\nabla r^{\boldsymbol{v}}(\boldsymbol{x}) = -\frac{\boldsymbol{v} - \langle \boldsymbol{v}, \boldsymbol{x} \rangle \boldsymbol{x}}{\sqrt{1 - \langle \boldsymbol{v}, \boldsymbol{x} \rangle^2}}, \tag{29}$$

$$\Delta r^{\boldsymbol{v}}(\boldsymbol{x}) = (d-1) \cot(r^{\boldsymbol{v}}(\boldsymbol{x})), \tag{30}$$

$$\left\langle \nabla r^{\boldsymbol{v}}(\boldsymbol{x}), \nabla r^{\boldsymbol{w}}(\boldsymbol{x}) \right\rangle = \frac{\langle \boldsymbol{v}, \boldsymbol{w} \rangle - \langle \boldsymbol{v}, \boldsymbol{x} \rangle \langle \boldsymbol{w}, \boldsymbol{x} \rangle}{\sqrt{(1 - \langle \boldsymbol{v}, \boldsymbol{x} \rangle^2)(1 - \langle \boldsymbol{w}, \boldsymbol{x} \rangle^2)}} = \frac{\langle \boldsymbol{v}, \boldsymbol{w} \rangle - \cos r^{\boldsymbol{v}}(\boldsymbol{x}) \cos r^{\boldsymbol{w}}(\boldsymbol{x})}{\sin r^{\boldsymbol{v}}(\boldsymbol{x}) \sin r^{\boldsymbol{w}}(\boldsymbol{x})}. \tag{31}$$

In particular, the logarithm map in Eq. (28) can be represented in radial distance:

$$\exp_{\boldsymbol{x}}^{-1}(\boldsymbol{v}) = -r^{\boldsymbol{v}}(\boldsymbol{x}) \nabla r^{\boldsymbol{v}}(\boldsymbol{x}), \tag{32}$$

### A.2 Connection Between Discrete Diffusion Models and Continuous Flow

In this section, we derive the connection between the discrete diffusion models and the continuous flow on a hypersphere.

**Continuous Flow on Hypersphere**  We first derive a useful lemma for continuous flows on hyperspheres. The following lemma describes a continuous flow on the hypersphere as a spherical linear interpolation.

---

**Lemma A.1.** Define a flow $\{Y_t\}_{t=0}^T$ on $\mathbb{S}^{d-1}$ from $\boldsymbol{y}_0 \in \mathbb{S}^{d-1}$ to $\boldsymbol{y}_1 \in \mathbb{S}^{d-1} \setminus \{\boldsymbol{y}_0, -\boldsymbol{y}_0\}$:

$$\frac{d\boldsymbol{Y}_t}{dt} = -\frac{d \log \kappa_t}{dt} \exp_{\boldsymbol{Y}_t}^{-1}(\boldsymbol{y}_1), \quad \boldsymbol{Y}_0 = \boldsymbol{y}_0, \tag{33}$$

where $\kappa_t : [0, T] \to [0, 1]$ is a scalar function satisfying $\kappa_0 = 1$ and $\kappa_T = 0$. Then the flow $\boldsymbol{Y}_t$ has a closed form solution:

$$\boldsymbol{Y}_t = \frac{\sin(\theta_0 - \theta_t)}{\sin \theta_0} \boldsymbol{y}_1 + \frac{\sin \theta_t}{\sin \theta_0} \boldsymbol{y}_0, \quad \theta_t := \kappa_t \cos^{-1} \langle \boldsymbol{y}_0, \boldsymbol{y}_1 \rangle, \tag{34}$$

which corresponds to the spherical linear interpolation, i.e., slerp:

$$\boldsymbol{Y}_t = \exp_{\boldsymbol{y}_1} \left( \kappa_t \exp_{\boldsymbol{y}_1}^{-1}(\boldsymbol{y}_0) \right) \tag{35}$$

---

*Proof.* Let $\theta_t := \cos^{-1} \langle \boldsymbol{Y}_t, \boldsymbol{y}_1 \rangle$. Then $\boldsymbol{Y}_t$ can be written as follows:

$$\boldsymbol{Y}_t = \cos \theta_t \boldsymbol{y}_1 + \sin \theta_t \boldsymbol{w}_t, \tag{36}$$

where $\boldsymbol{w}_t \in \mathbb{R}^d$ is an unit vector. From the definition of $\theta_t$, we have the following identity:

$$\frac{d\theta_t}{dt} = -\frac{1}{\sin \theta_t} \left\langle \frac{d\boldsymbol{Y}_t}{dt}, \boldsymbol{y}_1 \right\rangle = -\frac{1}{\sin \theta_t} \left\langle -\frac{d \log \kappa_t}{dt} \frac{\theta_t (\boldsymbol{y}_1 - \boldsymbol{Y}_t \cos \theta_t)}{\sin \theta_t}, \boldsymbol{y}_1 \right\rangle \tag{37}$$

$$= \frac{1}{\sin \theta_t} \frac{d \log \kappa_t}{dt} \theta_t \frac{1 - \cos^2 \theta_t}{\sin \theta_t} = \frac{d \log \kappa_t}{dt} \theta_t, \tag{38}$$

which yields representation of the flow $\boldsymbol{Y}_t$ in Eq. (33) with respect to $\theta$:

$$\frac{d\boldsymbol{Y}_t}{dt} = \frac{d\theta_t}{dt} \frac{\boldsymbol{y}_1 - \boldsymbol{Y}_t \cos \theta_t}{\sin \theta_t}. \tag{39}$$

Using the result of Eq. (39), we can see that $\boldsymbol{w}_t$ is a constant vector independent of $t$:

$$\frac{d\boldsymbol{w}_t}{dt} = \frac{1}{\sin^2 \theta_t} \left[ \left( \frac{d\boldsymbol{Y}_t}{dt} - \frac{d \cos \theta_t}{dt} \boldsymbol{y}_1 \right) \sin \theta_t - (\boldsymbol{Y}_t - \cos \theta_t \boldsymbol{y}_1) \frac{d \sin \theta_t}{dt} \right] \tag{40}$$

$$= \frac{1}{\sin^2 \theta_t} \frac{d\theta_t}{dt} \left[ -(\boldsymbol{y}_1 - \boldsymbol{Y}_t \cos \theta_t) + \sin^2 \theta_t \boldsymbol{y}_1 - \cos \theta_t \boldsymbol{Y}_t + \cos^2 \theta_t \boldsymbol{y}_1 \right] = 0. \tag{41}$$

Therefore we get the closed form solution for $\boldsymbol{Y}_t$:

$$\boldsymbol{Y}_t = \cos\theta_t \boldsymbol{y}_1 + \sin\theta_t \frac{\boldsymbol{y}_0 - \cos\theta_0 \boldsymbol{y}_1}{\sin\theta_0} = \frac{\sin(\theta_0 - \theta_t)}{\sin\theta_0}\boldsymbol{y}_1 + \frac{\sin\theta_t}{\sin\theta_0}\boldsymbol{y}_0, \tag{42}$$

where $\theta_t = \kappa_t \theta_0$ from Eq. (38). Note that the solution Eq. (34) is well-defined in the sense that $\sin\theta_0 > 0$ always holds. This is because $\|\langle \boldsymbol{Y}_t, \boldsymbol{y}_1\rangle\| \le 1$ as $\boldsymbol{Y}_t$ and $\boldsymbol{y}_1$ are on $\mathbb{S}^{d-1}$. Finally, using the definition of $\theta_t$, we can show the following:

$$\exp_{\boldsymbol{Y}_T}^{-1}(\boldsymbol{Y}_t) = \theta_t \frac{\boldsymbol{Y}_t - \boldsymbol{Y}_T \cos\theta_t}{\sin\theta_t} = \kappa_t \theta_0 \boldsymbol{w}_t = \kappa_t \theta_0 \boldsymbol{w}_0 = \kappa_t \exp_{\boldsymbol{Y}_T}^{-1}(\boldsymbol{Y}_0), \tag{43}$$

which gives the spherical linear interpolation defined in Eq. (35). $\qquad\square$

Our key observation is that the transition distribution $q_t(x_t|x)$ of a discrete diffusion process (Eq. (2)) is a categorical. Therefore, modeling $q_t$ is equivalent to modeling the continuous flow on the statistical manifold $\mathcal{P}(\mathcal{X})$. Here, we show that discrete diffusion models over $\mathcal{X}$ can be modeled by a continuous flow on $\mathbb{S}_+^{d-1}$. Specifically, we derive that the transition distribution of discrete diffusion processes can be modeled by the continuous flow on the hypersphere.

**Masked Diffusion Model**   We first show that discrete masked diffusion models correspond to a continuous flow on the statistical manifold starting from an absorbing state.

---

**Proposition A.2.** Define a flow $\{\boldsymbol{Y}_t\}_{t=0}^T$ on $\mathbb{S}^{d-1}$ from $\boldsymbol{e}_k$ to $\boldsymbol{e}_m$:

$$\frac{\mathrm{d}\boldsymbol{Y}_t}{\mathrm{d}t} = -\frac{\mathrm{d}\log\kappa_t}{\mathrm{d}t}\exp_{\boldsymbol{Y}_t}^{-1}(\boldsymbol{e}_m), \quad \boldsymbol{Y}_0 = \boldsymbol{e}_k, \quad \kappa_t = \frac{2}{\pi}\sin^{-1}(\sqrt{\alpha_t}) \tag{44}$$

where $\boldsymbol{e}_m$ denotes the absorbing state (i.e., mask state) and $\alpha_t \in [0,1]$ is some differentiable noise schedule satisfying $\alpha_0 \approx 1$ and $\alpha_1 \approx 0$. Then the random variable $\boldsymbol{Z}_t := \pi(\boldsymbol{Y}_t) \in \mathbb{R}^d$ satisfies the following:

$$\boldsymbol{Z}_t = \alpha_t \boldsymbol{e}_k + (1-\alpha_t)\boldsymbol{e}_m, \tag{45}$$

which is a flow that interpolates $\boldsymbol{e}_k$ and $\boldsymbol{e}_m$ on the probability simplex $\Delta^{d-1}$.

---

*Proof.* Using Lemma A.1 with $\theta_0 = \cos^{-1}\langle\boldsymbol{e}_m, \boldsymbol{e}_k\rangle = \pi/2$, we have the representation of $\boldsymbol{Y}_t$:

$$\boldsymbol{Y}_t = \sin(\theta_0 - \theta_t)\boldsymbol{e}_m + \sin\theta_t \boldsymbol{e}_k = \sqrt{1-\alpha_t}\boldsymbol{e}_m + \sqrt{\alpha_t}\boldsymbol{e}_k, \tag{46}$$

since $\theta_t = \sin^{-1}(\sqrt{\alpha_t})$. Therefore, $\boldsymbol{Z}_t$ has the following closed form:

$$\boldsymbol{Z}_t = (1-\alpha_t)\boldsymbol{e}_m + \alpha_t \boldsymbol{e}_k, \tag{47}$$

which defines a flow that interpolates $\boldsymbol{e}_k$ and $\boldsymbol{e}_m$ on the probability simplex $\Delta^{d-1}$. $\qquad\square$

Note that $\boldsymbol{Z}_t$ is a random variable on $\Delta^{d-1}$ representing the categorical distribution $\mathrm{Cat}(\alpha_t \boldsymbol{e}_{x_0} + (1-\alpha_t)\boldsymbol{e}_m)$. This corresponds to the transition distribution $q(x_t|x_0)$ of a discrete masked diffusion model, where the transition matrix for the diffusion process is given as follows:

$$Q_t^{absorb} = \begin{bmatrix} \alpha_t & 0 & \cdots & 0 & 0 \\ 0 & \alpha_t & \cdots & 0 & 0 \\ \vdots & \vdots & \ddots & \vdots & \vdots \\ 0 & 0 & \cdots & \alpha_t & 0 \\ 1-\alpha_t & 1-\alpha_t & \cdots & 1-\alpha_t & 0 \end{bmatrix} \tag{48}$$

---

**Corollary A.3.** The discrete masked diffusion process can be modeled by a continuous flow on $\mathbb{S}^{d-1}$ that starts from the absorbing state $\boldsymbol{e}_m$.

---

**Uniform Diffusion Model**   We also show that discrete uniform diffusion models correspond to a continuous flow on the statistical manifold that starts from the barycenter of the simplex.

**Proposition A.4.** Define a flow $\{Y_t\}_{t=0}^T$ on $\mathbb{S}^{d-1}$ from $e_k$ to $\sum_{i=1}^d e_i/\sqrt{d}$:

$$\frac{\mathrm{d}Y_t}{\mathrm{d}t} = -\frac{\mathrm{d}\log\kappa_t}{\mathrm{d}t}\exp_{Y_t}^{-1}\left(\sum_{i=1}^d \frac{1}{\sqrt{d}}e_i\right), \quad Y_0 = e_k, \tag{49}$$

$$\kappa_t = 1 - \frac{\sin^{-1}\left(\sqrt{1-\alpha_t}\sin\theta_0\right)}{\theta_0}, \quad \theta_0 := \cos^{-1}\left(\frac{1}{\sqrt{d}}\right) \tag{50}$$

where $\alpha_t \in [0,1]$ is a differentiable noise schedule satisfying $\alpha_0 \approx 1$ and $\alpha_1 \approx 0$. Then the random variable $Z_t := \pi(Y_t) \in \mathbb{R}^d$ satisfies the following:

$$Z_t = \sum_{i\neq k} \frac{1-\alpha_t}{d}e_i + \frac{1+(d-1)\alpha_t}{d}e_k, \tag{51}$$

which is a flow that interpolates $e_k$ and $\sum_{i=1}^d e_i/\sqrt{d}$ on the probability simplex $\Delta^{d-1}$.

*Proof.* Using Lemma A.1 with $\theta_0 = \cos^{-1}(1/\sqrt{d})$, we have the following representation of $Y_t$:

$$Y_t = \frac{\sin(\theta_0 - \theta_t)}{\sin\theta_0}\sum_{i=1}^d \frac{1}{\sqrt{d}}e_i + \frac{\sin\theta_t}{\sin\theta_0}e_k \tag{52}$$

$$= \sum_{i\neq k}\frac{\sin(\theta_0-\theta_t)}{\sqrt{d-1}}e_i + \left(\frac{\sqrt{d}\sin\theta_t}{\sqrt{d-1}} + \frac{\sin(\theta_0-\theta_t)}{\sqrt{d-1}}\right)e_k. \tag{53}$$

Due to the definition of $\kappa_t$, $Z_t$ has the following closed form:

$$Z_t = \sum_{i\neq k}\frac{1-\alpha_t}{d}e_i + \frac{1+(d-1)\alpha_t}{d}e_k, \tag{54}$$

which defines a flow that interpolates $e_k$ and $\sum_{i=1}^d e_i/\sqrt{d}$, i.e., the barycenter of the probability simplex $\Delta^{d-1}$. $\square$

Note that $Z_t$ is a random variable on $\Delta^{d-1}$ representing the categorical distribution:

$$\mathrm{Cat}\left(\sum_{i\neq x_0}\frac{1-\alpha_t}{d}e_i + \frac{1-(d-1)\alpha}{d}e_{x_0}\right), \tag{55}$$

which corresponds to the transition distribution $q(x_t|x_0)$ of a discrete uniform diffusion model. The transition matrix for the uniform diffusion process is given as follows:

$$Q^{unif} = \begin{bmatrix} 1-N & 1 & \cdots & 1 \\ 1 & 1-N & \cdots & 1 \\ \vdots & \vdots & \ddots & \vdots \\ 1 & 1 & \cdots & 1-N \end{bmatrix} \tag{56}$$

**Corollary A.5.** The discrete uniform diffusion process can be modeled by a continuous flow on $\mathbb{S}^{d-1}$ that starts from the barycenter of the probability simplex.

### A.3 Generative Process on Hypersphere

On a general manifold $\mathcal{M}$ that is complete, orientable, connected, and boundaryless, the logarithm bridge process [33] from $\boldsymbol{x}_0 \in \mathcal{M}$ to $\boldsymbol{x}_1 \in \mathcal{M}$ is defined as follows:

$$\mathrm{d}\bar{\boldsymbol{X}}_t = \gamma_t \exp^{-1}_{\bar{\boldsymbol{X}}_t}(\boldsymbol{x}_1)\mathrm{d}t + \sigma_t \mathrm{d}\mathbf{B}_t^{\mathcal{M}}, \quad \bar{\boldsymbol{X}}_0 = \boldsymbol{x}_0 \; ; \quad \gamma_t := \frac{\sigma_t^2}{\int_t^T \sigma_s^2 \mathrm{d}s} \tag{57}$$

where $\exp^{-1}_x(\cdot)$ denotes the logarithm map on $\mathcal{M}$ at point $x$ and $\mathbf{B}_t^{\mathcal{M}}$ is the Brownian motion defined on $\mathcal{M}$. In the case of $\mathcal{M} = \mathbb{S}^{d-1}$, we can derive the logarithm bridge process from $\boldsymbol{x}_0$ to $\boldsymbol{e}_k$:

$$\mathrm{d}\bar{\boldsymbol{X}}_t = \gamma_t \frac{\cos^{-1}\langle \bar{\boldsymbol{X}}_t, \boldsymbol{e}_k\rangle (\boldsymbol{e}_k - \langle \bar{\boldsymbol{X}}_t, \boldsymbol{e}_k\rangle \bar{\boldsymbol{X}}_t)}{\sqrt{1 - \langle \bar{\boldsymbol{X}}_t, \boldsymbol{e}_k\rangle^2}}\mathrm{d}t + \sigma_t \mathrm{d}\mathbf{B}_t^d, \quad \bar{\boldsymbol{X}}_0 = \boldsymbol{x}_0, \tag{58}$$

where we used the logarithm map of Eq. (28) and $\mathbf{B}_t^d$ is a Brownian motion defined on $\mathbb{S}^{d-1}$. It is worth noting that Eq. (58) is derived from applying the time change [64] to a simple bridge process:

$$\mathrm{d}\bar{\boldsymbol{X}}_t = \frac{1}{T-t}\frac{\cos^{-1}\langle \bar{\boldsymbol{X}}_t, \boldsymbol{e}_k\rangle (\boldsymbol{e}_k - \langle \bar{\boldsymbol{X}}_t, \boldsymbol{e}_k\rangle \bar{\boldsymbol{X}}_t)}{\sqrt{1 - \langle \bar{\boldsymbol{X}}_t, \boldsymbol{e}_k\rangle^2}}\mathrm{d}t + \mathrm{d}\mathbf{B}_t^d, \quad \bar{\boldsymbol{X}}_0 = \boldsymbol{x}_0. \tag{59}$$

Note that the drift of the logarithm bridge process can be rewritten using the geodesic distance $d_g(\cdot, \cdot)$ as follows:

$$\mathrm{d}\bar{\boldsymbol{X}}_t = \left[\gamma_t \cos^{-1}\langle \bar{\boldsymbol{X}}_t, \boldsymbol{e}_k\rangle \nabla_{\bar{\boldsymbol{X}}_t} d_g(\bar{\boldsymbol{X}}_t, \boldsymbol{e}_k)\right]\mathrm{d}t + \sigma_t \mathrm{d}\mathbf{B}_t^d, \quad \bar{\boldsymbol{X}}_0 = \boldsymbol{x}_0. \tag{60}$$

The direction of the drift corresponds to the direction that minimizes the distance between the current state $\bar{\boldsymbol{X}}_t$ and the endpoint $\boldsymbol{e}_k$. Since $\gamma_t \to \infty$ as $t \to T$, the bridge process converges to the endpoint $\boldsymbol{e}_k$. The convergence behavior can be analyzed by examining the radial process $r_t^k := d_g(\boldsymbol{e}_k, \boldsymbol{X}_t)$, which we describe below.

**Radial Process**  Let $r_t^{\boldsymbol{w}} := d_g(\boldsymbol{w}, \boldsymbol{X}_t)$ for arbitrary point $\boldsymbol{w} \in \mathbb{S}^{d-1}$. Then the bridge process of Eq. (58) can be rewritten as follows:

$$\mathrm{d}\bar{\boldsymbol{X}}_t = \gamma_t \frac{r_t^k(\boldsymbol{e}_k - \cos r_t^k \bar{\boldsymbol{X}}_t)}{\sin r_t^k}\mathrm{d}t + \sigma_t \mathrm{d}\mathbf{B}_t^d, \quad \boldsymbol{X}_0 = \boldsymbol{x}_0, \tag{61}$$

where $r_t^k := r_t^{\boldsymbol{e}_k}$. Then the SDE of $r_t^{\boldsymbol{w}}$ can be derived using the Itô's formula as follows:

$$\mathrm{d}r_t^{\boldsymbol{w}} = \left[\left\langle \nabla r_t^{\boldsymbol{w}}, \gamma_t \frac{r_t^k(\boldsymbol{e}_k - \cos r_t^k \bar{\boldsymbol{X}}_t)}{\sin r_t^k}\right\rangle + \frac{\sigma_t^2}{2}\Delta r_t^{\boldsymbol{w}}\right]\mathrm{d}t + \left\langle \nabla r_t^{\boldsymbol{w}}, \sigma_t \mathrm{d}\mathbf{B}_t^d\right\rangle, \tag{62}$$

where $\nabla$ and $\Delta$ denote the Riemannian gradient and the Laplace-Beltrami operator on $\mathbb{S}^{d-1}$, respectively. From the identities in Appendix A.1 and the fact that $\langle \nabla r_t^{\boldsymbol{w}}, \mathrm{d}\mathbf{B}_t^d\rangle$ is a 1-dimensional Brownian motion ([32] Example 3.3.3), we get the following result:

$$\mathrm{d}r_t^{\boldsymbol{w}} = \left[-\gamma_t\, r_t^k \frac{\langle \boldsymbol{e}_k, \boldsymbol{w}\rangle - \cos r_t^k \cos r_t^{\boldsymbol{w}}}{\sin r_t^k \sin r_t^{\boldsymbol{w}}} + \frac{(d-1)\sigma_t^2}{2}\cot(r_t^{\boldsymbol{w}})\right]\mathrm{d}t + \sigma_t \mathrm{d}W_t,$$

$$r_0^{\boldsymbol{w}} := \cos^{-1}\langle \boldsymbol{x}_0, \boldsymbol{w}\rangle, \tag{63}$$

where $W_t$ denotes a 1-dimensional Brownian motion. For $\boldsymbol{w} = \boldsymbol{e}_l$, we obtain a simplified formulation:

$$\mathrm{d}r_t^l = \left[-\gamma_t C(r_t^k, r_t^l)r_t^k + \frac{(d-1)\sigma_t^2}{2}\cot(r_t^l)\right]\mathrm{d}t + \sigma_t \mathrm{d}W_t, \quad r_0^l = \frac{\pi}{2}\delta_{k,l} \tag{64}$$

$$C(r_t^k, r_t^l) = \begin{cases} 1 & \text{if } k = l \\ -\cot(r_t^k)\cot(r_t^l) & \text{otherwise} \end{cases}. \tag{65}$$

## A.4 Diffusion Mixture Representation

We provide the statement of the diffusion mixture representation from Jo and Hwang [33], which extends Peluchetti [45] to Riemannian manifolds. We refer the readers to Jo and Hwang [33] for a detailed derivation of the diffusion mixture representation for general Riemannian manifolds. We consider Riemannian manifolds that are complete, orientable, connected, and boundaryless.

---

**Proposition A.6.** Consider a collection of SDEs on a manifold $\mathcal{M}$ indexed by $\lambda \in \Lambda$:

$$\mathrm{d}\boldsymbol{X}_t^\lambda = \eta^\lambda(\boldsymbol{X}_t^\lambda, t)\mathrm{d}t + \sigma^\lambda(\boldsymbol{X}_t^\lambda, t)\,\mathrm{d}\mathbf{B}_t^{\mathcal{M}}, \quad \boldsymbol{X}_0^\lambda \sim p_0 \tag{66}$$

with marginal distribution of $\boldsymbol{X}_t^\lambda$ denoted by $p_t^\lambda$. Let $\mathcal{L}$ be a mixing distribution over $\Lambda$. Then a diffusion process on $\mathcal{M}$ described by the SDE:

$$\mathrm{d}\boldsymbol{X}_t = \eta(\boldsymbol{X}_t, t)\mathrm{d}t + \sigma(\boldsymbol{X}_t, t)\,\mathrm{d}\mathbf{B}_t^{\mathcal{M}}, \quad \boldsymbol{X}_0 \sim p_0 \tag{67}$$

$$\eta(x, t) = \int \eta^\lambda(x, t)\frac{p_t^\lambda(x)}{p_t(x)}\mathcal{L}(\mathrm{d}\lambda) \,, \quad \sigma(x, t) = \left(\int a^\lambda(x, t)\frac{p_t^\lambda(x)}{p_t(x)}\mathcal{L}(\mathrm{d}\lambda)\right)^{1/2} \tag{68}$$

where $a^\lambda := \sigma^\lambda(\sigma^\lambda)^\top$, admits the marginal distribution $p_t$:

$$p_t(x) = \int p_t^\lambda(x)\mathcal{L}(\mathrm{d}\lambda), \quad p_0(x) = \int p_0^\lambda(x)\mathcal{L}(\mathrm{d}\lambda). \tag{69}$$

---

From the diffusion mixture representation, Jo and Hwang [33] construct the generative process as a mixture of the bridge processes on $\mathcal{M}$ as shown in the following proposition.

---

**Proposition A.7.** Let $p_0$ and $p_1$ be probability distributions on a Riemannian manifold $\mathcal{M}$. Consider a collection of SDEs that describes bridge processes on $\mathcal{M}$ from $x \sim p_0$ to $y \sim p_1$:

$$\mathrm{d}\boldsymbol{X}_t^{x,y} = \eta^{x,y}(\boldsymbol{X}_t^{x,y}, t)\mathrm{d}t + \sigma_t\mathrm{d}\mathbf{B}_t^{\mathcal{M}}, \quad \boldsymbol{X}_0 = x, \tag{70}$$

with marginal distribution of $\boldsymbol{X}^{x,y}$ denoted by $p_t^{x,y}$. Then the following SDE defines a diffusion process that transports an initial distribution $p_0$ to a target distribution $p_1$:

$$\mathrm{d}\boldsymbol{X}_t = \eta(\boldsymbol{X}_t, t)\mathrm{d}t + \sigma_t\mathbf{B}_t^{\mathcal{M}}, \quad \boldsymbol{X}_0 \sim p_0, \tag{71}$$

$$\eta(z, t) := \iint \eta^{x,y}(z, t)\frac{p_t^{x,y}(z)}{p_t(z)}p_0(\mathrm{dvol}_x)p_1(\mathrm{dvol}_y), \tag{72}$$

$$p_t(z) := \iint p_t^{x,y}(z)p_0(\mathrm{dvol}_x)p_1(\mathrm{dvol}_y). \tag{73}$$

---

In the case of $\mathcal{M} = \mathbb{S}^{d-1}$, we derive the generative process for the reparameterized data distribution $p_{data}(x) = \sum_{k=1}^d p_k\delta(x - \boldsymbol{e}_k)$, by mixing the logarithm bridge processes on $\mathbb{S}^{d-1}$ (Eq. (5)).

---

**Corollary A.8.** Let $p_{data}(x) = \sum_{k=1}^d p_k\delta(x - \boldsymbol{e}_k)$ be a data distribution on $\mathbb{S}^{d-1}$. Then the following SDE defines a diffusion process that transports the initial point $\boldsymbol{x}_0 \in \mathbb{S}^{d-1}$ to the distribution $p_{data}$:

$$\mathrm{d}\boldsymbol{X}_t = \left[\sum_{k=1}^d p_{T|t}(\boldsymbol{e}_k|\boldsymbol{X}_t)\,\eta^k(\boldsymbol{X}_t, t)\right]\mathrm{d}t + \sigma_t\mathrm{d}\mathbf{B}_t^d, \quad \boldsymbol{X}_0 = \boldsymbol{x}_0, \tag{74}$$

$$\eta^k(z, t) := \gamma_t\frac{\cos^{-1}\langle z, \boldsymbol{e}_k\rangle(\boldsymbol{e}_k - \langle z, \boldsymbol{e}_k\rangle z)}{\sqrt{1 - \langle z, \boldsymbol{e}_k\rangle^2}}, \tag{75}$$

where $p_{T|t}(\boldsymbol{e}_k|\boldsymbol{X}_t)$ represents the conditional probability that the process will reach the endpoint $\boldsymbol{e}_k$ at time $T$, given the current state $\boldsymbol{X}_t$ at time $t$.

---

## A.5 Mixture Paths

We derive a new family of generative processes by constructing a mixture over the time marginals of generative processes. We first present a proposition for mixing diffusion processes with a general time-dependent mixing schedule.

**Proposition A.9.** Consider a collection of $n$ SDEs on a closed Riemannian manifold $\mathcal{M}$:

$$\mathrm{d}\boldsymbol{X}_t^i = \eta^i(\boldsymbol{X}_t^i, t)\mathrm{d}t + \sigma^i(\boldsymbol{X}_t^i, t)\,\mathrm{d}\mathbf{B}_t^{\mathcal{M}}, \quad \boldsymbol{X}_0^i \sim p_0 \tag{76}$$

with marginal distribution of $\boldsymbol{X}_t^i$ denoted by $p_t^i$. Let $\lambda^i \in C^1([0,T])$ satisfy $\lambda_t^i \geq 0$ and $\sum_{i=1}^n \lambda_t^i = 1$ for all $t$. Then there exists a diffusion process with the marginal distribution $p_t$:

$$p_t(x) = \sum_{i=1}^n \lambda_t^i p_t^i(x). \tag{77}$$

*Proof.* We show that there exists a scalar potential $\Phi : \mathcal{M} \times [0,T] \to \mathbb{R}$ such that the following SDE defines a diffusion process that yields the desired marginal distribution:

$$\mathrm{d}\boldsymbol{X}_t = \eta(\boldsymbol{X}_t, t)\mathrm{d}t + \sigma(\boldsymbol{X}_t, t)\mathrm{d}\mathbf{B}_t^{\mathcal{M}}, \tag{78}$$

$$\eta(x,t) := \sum_{i=1}^n \lambda_t^i \eta^i(x,t)\frac{p_t^i(x)}{p_t(x)} - \frac{\nabla\Phi(x,t)}{p_t(x)} - \frac{1}{2}\sum_{i=1}^n \lambda_t^i a^i(x,t)\nabla\left(\frac{p_t^i(x)}{p_t(x)}\right) \tag{79}$$

$$\sigma(x,t) := \left(\sum_{i=1}^n \lambda_t^i a^i(x,t)\frac{p_t^i(x)}{p_t(x)}\right)^{1/2}, \tag{80}$$

where $a^i := \sigma^i(\sigma^i)^\top$. Here, we assume that $\eta^i$ and $\sigma^i$ are bounded and $a^i$ are uniformly elliptic.

First, define a function $f : \mathcal{M} \to \mathbb{R}$ that satisfies the zero-mean condition:

$$f(x,t) := \sum_{i=1}^n \frac{\mathrm{d}\lambda_t^i}{\mathrm{d}t}p_t^i(x) \; ; \; \int_{\mathcal{M}} f(x,t)\mathrm{dvol}_x = \sum_{i=1}^n \frac{\mathrm{d}\lambda_t^i}{\mathrm{d}t}\int_{\mathcal{M}} p_t^i(x)\mathrm{dvol}_x = \sum_{i=1}^n \frac{\mathrm{d}\lambda_t^i}{\mathrm{d}t} = 0, \tag{81}$$

where we used the fact that $\sum_{i=1}^n \lambda_t^i = 1$ for all $t$. As $\mathcal{M}$ is closed, its Laplace–Beltrami operator is invertible on the subspace of zero-mean functions. Therefore, the Poisson equation $\Delta\Phi(\cdot, t) = f(\cdot, t)$ admits a weak solution $\Phi$.

From the definition of $p_t$, we can derive the following equality:

$$\frac{\partial p_t(x)}{\partial t} = \sum_{i=1}^n \frac{\partial(\lambda_t^i p_t^i(x))}{\partial t} = \sum_{i=1}^n \lambda_t^i \frac{\partial p_t^i(x)}{\partial t} + \sum_{i=1}^n \frac{\mathrm{d}\lambda_t^i}{\mathrm{d}t}p_t^i(x) \tag{82}$$

$$= \sum_{i=1}^n \lambda_t^i\left[-\mathrm{div}\left(p_t^i(x)\eta^i(x,t)\right) + \frac{1}{2}\mathrm{div}\left(a^i(x,t)\nabla p_t^i(x)\right)\right] + \Delta\Phi(x,t) \tag{83}$$

$$= -\mathrm{div}\left(\sum_{i=1}^n \lambda_t^i p_t^i(x)\eta^i(x,t)\right) + \frac{1}{2}\sum_{i=1}^n \lambda_t^i\mathrm{div}\left(a^i(x,t)\nabla p_t^i(x)\right) + \mathrm{div}(\nabla\Phi(x,t)) \tag{84}$$

$$\begin{aligned}= &-\mathrm{div}\left(\sum_{i=1}^n \lambda_t^i p_t^i(x)\eta^i(x,t) - \nabla\Phi(x,t)\right) \\ &+ \frac{1}{2}\sum_{i=1}^n \mathrm{div}\left(a^i(x,t)\left[\nabla p_t(x)\frac{\lambda_t^i p_t^i(x)}{p_t(x)} + p_t(x)\lambda_t^i\nabla\left(\frac{p_t^i(x)}{p_t(x)}\right)\right]\right)\end{aligned} \tag{85}$$

where we used the product rule for divergence in $\lambda_t^i p_t^i(x) = p_t(x)\frac{\lambda_t^i p_t^i(x)}{p_t(x)}$.

Reordering the terms in Eq. (85), we obtain the following result:

$$\frac{\partial p_t(x)}{\partial t} = -\text{div}\left(p_t(x)\left[\sum_{i=1}^n \lambda_t^i \eta^i(\boldsymbol{X}_t, t)\frac{p_t^i(\boldsymbol{X}_t)}{p_t(\boldsymbol{X}_t)} - \frac{\nabla\Phi(\boldsymbol{X}_t, t)}{p_t(\boldsymbol{X}_t)} - \frac{1}{2}\sum_{i=1}^n \lambda_t^i a^i(\boldsymbol{X}_t, t)\nabla\left(\frac{p_t^i(x)}{p_t(x)}\right)\right]\right)$$
$$+ \frac{1}{2}\text{div}\left(\left[\sum_{i=1}^n \lambda_t^i a^i(\boldsymbol{X}_t, t)\frac{p_t^i(\boldsymbol{X}_t)}{p_t(\boldsymbol{X}_t)}\right]\nabla p_t(x)\right), \tag{86}$$

which corresponds to the Fokker-Planck equation for the SDE of Eq. (78). Therefore, the diffusion process described by the SDE in Eq. (78) has a marginal distribution $p_t$ in Eq. (77). $\qquad\square$

From Proposition A.9, we can derive a new family of generative processes by constructing a mixture over the time marginals of generative processes $\{\mathbb{Q}^i : 1 \le i \le n\}$:

$$\mathbb{Q}_t^{mix} \coloneqq \sum_{i=1}^n \lambda_t^i \mathbb{Q}_t^i \ , \ \ \sum_{i=1}^n \lambda_t^i = 1, \ 0 \le \lambda_t^i \le 1, \tag{87}$$

where $\lambda_t^i$ is the time-dependent mixing schedule assigned to the $i$-the generative path.

One example is creating a mixture path by mixing the masked diffusion and the uniform diffusion on $\mathbb{S}^{d-1}$, as defined in Section 3.2.

> **Corollary A.10.** Let $p_t^{mask}$ and $p_t^{unif}$ denote the marginal distributions of the masked diffusion and the uniform diffusion on $\mathbb{S}^{d-1}$, as defined in Section 3.2, respectively. Then there exists a diffusion process on $\mathbb{S}^{d-1}$ whose marginal distribution at time $t$ satisfies:
>
> $$p_t(x) = \lambda_t p_t^{mask}(x) + (1 - \lambda_t)p_t^{unif}(x), \tag{88}$$
>
> where $\lambda_t \in [0, 1]$ for all $t \in [0, T]$.

## A.6 Likelihood Bound

We derive the point-wise likelihood bound and the upper bound on the negative log-likelihood of our generative model, defined as the parameterized mixture process $\mathbb{Q}^\theta$ with the drift $\eta_\theta$ in Eq. (11).

Let $\mathbb{Q}^k$ be a bridge process with starting point $\boldsymbol{x}_0$ and endpoint $\boldsymbol{e}_k$. From the KL divergence between $\mathbb{Q}^\theta$ and $\mathbb{Q}^k$, we can derive a point-wise upper bound on the negative log-likelihood using the Girsanov theorem on compact manifolds (De Bortoli et al. [16], Corollary H.3):

$$-\log \hat{p}_\theta(\boldsymbol{e}_k) = D_{KL}(\delta(\boldsymbol{e}_k)\|\hat{p}_\theta(\boldsymbol{e}_k)) = D_{KL}(\mathbb{Q}_T^k\|\mathbb{Q}_T^\theta) \tag{89}$$

$$\le D_{KL}(\mathbb{Q}^k\|\mathbb{Q}^\theta) = \mathbb{E}_{\boldsymbol{X}\sim\mathbb{Q}^k}\left[\frac{1}{2}\int_0^T \left\|\sigma_t^{-1}\big(\eta_\theta(\boldsymbol{X}_t, t) - \eta^k(\boldsymbol{X}_t, t)\big)\right\|_2^2 dt\right], \tag{90}$$

where the inequality comes from the data-processing inequality. The point-wise likelihood bound leads to the upper bound on the negative likelihood of our model:

$$\mathbb{E}_{\boldsymbol{z}\sim p_{data}}\big[-\log\hat{p}_\theta(\boldsymbol{z})\big] \le \mathbb{E}_{\substack{\boldsymbol{e}_k\sim p_{data}\\\boldsymbol{X}\sim\mathbb{Q}^k}}\left[\frac{1}{2}\int_0^T \left\|\sigma_t^{-1}\big(\eta_\theta(\boldsymbol{X}_t, t) - \eta^k(\boldsymbol{X}_t, t)\big)\right\|_2^2 dt\right]. \tag{91}$$

## A.7 Training Objective

We show that minimizing the cross-entropy-based loss defined in Eq. (15) guarantees maximizing the likelihood of our generative model defined as the parameterized mixture process in Eq. (11).

We start with deriving a uniform bound for the drift of the bridge process defined in Eq. (5):

$$\left\|\eta^l(z, t)\right\|_2 = \left\|\gamma_t\frac{\cos^{-1}\langle z, \boldsymbol{e}_l\rangle(\boldsymbol{e}_l - \langle z, \boldsymbol{e}_l\rangle z)}{\sqrt{1 - \langle z, \boldsymbol{e}_l\rangle^2}}\right\|_2 = \gamma_t\cos^{-1}\langle z, \boldsymbol{e}_l\rangle \le \pi\gamma_t. \tag{92}$$

Then the triangle inequality gives the following:

$$\left\|\sum_{l=1}^{d}\langle p_\theta(x,t),\boldsymbol{e}_l\rangle\eta^l(x,t)-\eta^k(x,t)\right\|_2^2 \le \left(\sum_{l=1}^{d}\left|\langle p_\theta(x,t),\boldsymbol{e}_l\rangle-\delta_{k,l}\right|\|\eta^l(x,t)\|_2\right)^2 \tag{93}$$

$$\le \pi^2\gamma_t^2\left(\sum_{l=1}^{d}\left|\langle p_\theta(x,t),\boldsymbol{e}_l\rangle-\delta_{k,l}\right|\right)^2 \le -2\pi^2\gamma_t^2\log\langle p_\theta(x,t),\boldsymbol{e}_k\rangle. \tag{94}$$

From Eq. (94), we derive the upper bound for the maximum likelihood training objective $\mathcal{L}(\theta)$ in Eq. (14) as follows:

$$\mathcal{L}(\theta) = \mathbb{E}_{\substack{\boldsymbol{e}_k\sim p_{data}\\ \boldsymbol{X}\sim\mathbb{Q}^k}}\left[\frac{1}{2}\int_0^T \sigma_t^{-2}\left\|\sum_{l=1}^{d}\langle p_\theta(\boldsymbol{X}_t,t),\boldsymbol{e}_l\rangle\eta^l(\boldsymbol{X}_t,t)-\eta^k(\boldsymbol{X}_t,t)\right\|_2^2 dt\right] \tag{95}$$

$$\le \mathbb{E}_{\substack{\boldsymbol{e}_k\sim p_{data}\\ \boldsymbol{X}\sim\mathbb{Q}^k}}\left[\int_0^T -\frac{2\pi^2\gamma_t^2}{\sigma_t^2}\log\langle p_\theta(\boldsymbol{X}_t,t),\boldsymbol{e}_k\rangle dt\right] \tag{96}$$

$$\le \mathbb{E}_{\substack{\boldsymbol{e}_k\sim p_{data}\\ \boldsymbol{X}\sim\mathbb{Q}^k}}\left[\left(\sup_{t\in[0,T-\epsilon]}\frac{2\pi^2\gamma_t^2}{\sigma_t^2}\right)\int_0^{T-\epsilon} -\log\langle p_\theta(\boldsymbol{X}_t,t),\boldsymbol{e}_k\rangle dt\right]$$
$$+ \mathbb{E}_{\substack{\boldsymbol{e}_k\sim p_{data}\\ \boldsymbol{X}\sim\mathbb{Q}^k}}\left[\int_{T-\epsilon}^T -\frac{2\pi^2\gamma_t^2}{\sigma_t^2}\log\langle p_\theta(\boldsymbol{X}_t,t),\boldsymbol{e}_k\rangle dt\right] \tag{97}$$

$$\le M_\epsilon\mathcal{L}^{CE}(\theta)+F(\epsilon), \tag{98}$$

where $F(\epsilon)$ denotes the last term of Eq. (97). Since $\boldsymbol{X}\sim\mathbb{Q}^k$ is the bridge process with endpoint $\boldsymbol{e}_k$, $\boldsymbol{X}_t$ converges to $\boldsymbol{e}_k$ as $t\to T$ and $\langle p_\theta(\boldsymbol{X}_{T-\epsilon},T-\epsilon),\boldsymbol{e}_k\rangle\approx 1$ for sufficiently small $\epsilon>0$. As a result, $F(\epsilon)\approx 0$ for sufficiently small $\epsilon$, which lead to the following result:

$$\mathcal{L}(\theta)\le M\mathcal{L}^{CE}(\theta), \tag{99}$$

for some constant $M>0$. Therefore, minimizing the cross-entropy-based loss $\mathcal{L}^{CE}(\theta)$ approximately guarantees maximizing the likelihood.

## A.8  Projected Processes

Let $\boldsymbol{X}_{t|0,T}$ denote the mixture process $\{\boldsymbol{X}_t\}_{t=0}^T$ on $\mathbb{S}^{d-1}$ conditioned to the endpoints $\boldsymbol{X}_0=\boldsymbol{x}_0$ and $\boldsymbol{X}_T=\boldsymbol{x}_1$. Then $\boldsymbol{X}_{t|0,T}$ corresponds to a bridge process described by the following SDE:

$$d\bar{\boldsymbol{X}}_t = \gamma_t\frac{\cos^{-1}\langle\bar{\boldsymbol{X}}_t,\boldsymbol{x}_1\rangle(\boldsymbol{x}_1-\langle\bar{\boldsymbol{X}}_t,\boldsymbol{x}_1\rangle\bar{\boldsymbol{X}}_t)}{\sqrt{1-\langle\bar{\boldsymbol{X}}_t,\boldsymbol{x}_1\rangle^2}}dt+\sigma_t d\mathbf{B}_t^d,\ \bar{\boldsymbol{X}}_0=\boldsymbol{x}_0. \tag{100}$$

We can derive the projection $z_t^T=\langle\boldsymbol{X}_{t|0,T},\boldsymbol{x}_1\rangle$ using the Itô's formula for $f_T(\cdot):=\langle\cdot,\boldsymbol{x}_1\rangle$:

$$dz_t^T = \left[\left\langle\nabla f_T(\bar{\boldsymbol{X}}_t),\gamma_t\frac{\cos^{-1}\langle\bar{\boldsymbol{X}}_t,\boldsymbol{x}_1\rangle(\boldsymbol{x}_1-\langle\bar{\boldsymbol{X}}_t,\boldsymbol{x}_1\rangle\bar{\boldsymbol{X}}_t)}{\sqrt{1-\langle\bar{\boldsymbol{X}}_t,\boldsymbol{x}_1\rangle^2}}\right\rangle+\frac{1}{2}\sigma_t^2\Delta f_T(\bar{\boldsymbol{X}}_t)\right]dt$$
$$+\sigma_t\left\langle\nabla f_T(\bar{\boldsymbol{X}}_t),d\mathbf{B}_t^d\right\rangle \tag{101}$$

$$= \left[\left\langle\boldsymbol{x}_1-\langle\bar{\boldsymbol{X}}_t,\boldsymbol{x}_1\rangle\bar{\boldsymbol{X}}_t,\gamma_t\frac{\cos^{-1}z_t^T}{\sqrt{1-(z_t^T)^2}}\left(\boldsymbol{x}_1-\langle\bar{\boldsymbol{X}}_t,\boldsymbol{x}_1\rangle\bar{\boldsymbol{X}}_t\right)\right\rangle-\frac{(d-1)\sigma_t^2}{2}z_t^T\right]dt$$
$$+\sigma_t\sqrt{1-(z_t^T)^2}dW_t \tag{102}$$

$$= \left[\gamma_t\cos^{-1}z_t^T\sqrt{1-(z_t^T)^2}-\frac{(d-1)\sigma_t^2}{2}z_t^T\right]dt+\sigma_t\sqrt{1-(z_t^T)^2}dW_t, \tag{103}$$

where we have used the identities $\nabla f_T(\boldsymbol{z}) = \boldsymbol{x}_1 - \langle \boldsymbol{z}, \boldsymbol{x}_1 \rangle \boldsymbol{z}, \Delta f_T(\boldsymbol{z}) = -(d-1)\langle \boldsymbol{z}, \boldsymbol{x}_1 \rangle$. Note that the Laplace-Beltrami operator defined on $\mathbb{S}^{d-1}$ has a simple and tractable form due to the radial symmetry of the hypersphere.

Similarly, $z_t^0 = \langle \bar{\boldsymbol{X}}_t, \boldsymbol{x}_0 \rangle$ can be derived using Itô's formula for $f_0(\boldsymbol{z}) := \langle \boldsymbol{z}, \boldsymbol{x}_0 \rangle$:

$$\mathrm{d}z_t^0 = \left[ \gamma_t \frac{\cos^{-1} z_t^T}{\sqrt{1 - (z_t^T)^2}} \left( \langle \boldsymbol{x}_0, \boldsymbol{x}_1 \rangle - z_t^0 z_t^T \right) - \frac{(d-1)\sigma_t^2}{2} z_t^0 \right] \mathrm{d}t + \sigma_t \sqrt{1 - (z_t^0)^2} \mathrm{d}W_t. \quad (104)$$

**Masked Diffusion** Since the masked bridge process has $\boldsymbol{x}_0 = \boldsymbol{e}_m$ and $\boldsymbol{x}_1 = \boldsymbol{e}_k$ with $\langle \boldsymbol{e}_m, \boldsymbol{e}_k \rangle = 0$ for all non-mask token $\boldsymbol{e}_k$, the projected processes are described as the follows:

$$\mathrm{d}z_t^l = \left[ \gamma_t \frac{\cos^{-1} z_t^k}{\sqrt{1 - (z_t^k)^2}} \left( \delta_{l,k} - z_t^l z_t^k \right) - \frac{(d-1)\sigma_t^2}{2} z_t^l \right] \mathrm{d}t + \sigma_t \sqrt{1 - (z_t^l)^2} \mathrm{d}W_t^l, \quad (105)$$

with initial condition $z_0^l = 0$ for all $l$ and $W_t^l$ are 1-dimensional standard Wiener processes.

**Uniform Diffusion** The uniform bridge process has $\boldsymbol{x}_0 = \sum_{i=1}^d \boldsymbol{e}_i / \sqrt{d}$ and $\boldsymbol{x}_1 = \boldsymbol{e}_k$, and the projected processes have a simple form:

$$\mathrm{d}z_t^l = \left[ \gamma_t \frac{\cos^{-1} z_t^k}{\sqrt{1 - (z_t^k)^2}} \left( A_{l,k} - z_t^l z_t^k \right) - \frac{(d-1)\sigma_t^2}{2} z_t^l \right] \mathrm{d}t + \sigma_t \sqrt{1 - (z_t^l)^2} \mathrm{d}W_t^l,$$

$$A_{l,k} = \begin{cases} 1/\sqrt{d} & \text{if } l \neq k \\ 1 & \text{otherwise} \end{cases} \quad (106)$$

with initial condition $z_0^l = 1/\sqrt{d}$ for all $l$.

### A.9 Simulation-Free Training with Radial Symmetry

Here we derive the parameters of the Riemannian normal distribution from the projected processes:

$$\mathrm{d}z_t^T = \left[ \gamma_t \cos^{-1} z_t^T \sqrt{1 - (z_t^T)^2} - \frac{(d-1)\sigma_t^2}{2} z_t^T \right] \mathrm{d}t + \sigma_t \sqrt{1 - (z_t^T)^2} \, \mathrm{d}W_t^T, \quad (107)$$

$$\mathrm{d}z_t^0 = \left[ \gamma_t \frac{\cos^{-1} z_t^T}{\sqrt{1 - (z_t^T)^2}} \left( z_0^T - z_t^0 z_t^T \right) - \frac{(d-1)\sigma_t^2}{2} z_t^0 \right] \mathrm{d}t + \sigma_t \sqrt{1 - (z_t^0)^2} \, \mathrm{d}W_t^0, \quad (108)$$

with initial conditions $z_0^T = \langle \boldsymbol{X}_0, \boldsymbol{X}_T \rangle$ and $z_0^0 = 1$. From the definition $z_t^T := \langle \boldsymbol{X}_{t|0,T}, \boldsymbol{x}_1 \rangle$, we establish the connection between the mean projection $\mathbb{E}z_t^T$ and the parameters $\alpha_t$ and $\rho_t$:

$$\mathbb{E}z_t^T \approx \mathbb{E}_{\boldsymbol{z}} \langle \exp_{\boldsymbol{\mu}_t}(\rho_t \boldsymbol{z}), \boldsymbol{x}_1 \rangle, \quad \boldsymbol{z} \sim \mathcal{N}_{T_{\boldsymbol{\mu}_t}\mathbb{S}^d}(\boldsymbol{0}, \mathbf{I}) \quad (109)$$

$$\overset{\text{Eq. (28)}}{=} \mathbb{E}_{\boldsymbol{z}} \left\langle \cos(\rho_t\|\boldsymbol{z}\|)\boldsymbol{\mu}_t + \sin(\rho_t\|\boldsymbol{z}\|)\frac{\boldsymbol{z}}{\|\boldsymbol{z}\|}, \boldsymbol{x}_1 \right\rangle \quad (110)$$

$$= \mathbb{E}_{\boldsymbol{z}} \left( \cos(\rho_t\|\boldsymbol{z}\|) \langle \boldsymbol{\mu}_t, \boldsymbol{x}_1 \rangle \right) + \underbrace{\mathbb{E}_{\boldsymbol{z}} \left( \sin(\rho_t\|\boldsymbol{z}\|) \left\langle \frac{\boldsymbol{z}}{\|\boldsymbol{z}\|}, \boldsymbol{x}_1 \right\rangle \right)}_{=0} \quad (111)$$

$$\overset{\text{Eq. (17)}}{=} \mathbb{E}_{\boldsymbol{z}} \cos(\rho_t\|\boldsymbol{z}\|) \left\langle \frac{\alpha_t}{\sin\phi_0}\boldsymbol{x}_1 + \left( \sqrt{1 - \alpha_t^2} - \frac{\alpha_t \cos\phi_0}{\sin\phi_0} \right) \boldsymbol{x}_0, \boldsymbol{x}_1 \right\rangle \quad (112)$$

$$= \mathbb{E}_{\boldsymbol{z}} \cos(\rho_t\|\boldsymbol{z}\|) \left( \sin\phi_0 \alpha_t + \cos\phi_0 \sqrt{1 - \alpha_t^2} \right), \quad (113)$$

for $\phi_0 := \cos^{-1}\langle \boldsymbol{X}_0, \boldsymbol{X}_T \rangle$, where the last term in Eq. (111) is zero due to radial symmetry. Similarly,

$$\mathbb{E}z_t^0 \approx \mathbb{E}_{\boldsymbol{z}} \langle \exp_{\boldsymbol{\mu}_t}(\rho_t \boldsymbol{z}), \boldsymbol{x}_0 \rangle = \mathbb{E}_{\boldsymbol{z}} \cos(\rho_t\|\boldsymbol{z}\|) \sqrt{1 - \alpha_t^2}, \quad (114)$$

Notably, we have the following identity for $z \sim \mathcal{N}_{T_{\mu_t} \mathbb{S}^d}(\mathbf{0}, \mathbf{I})$:

$$\mathbb{E}_z \cos(\rho_t \|z\|) = e^{-\rho_t^2/2} {}_1f_1\left(\frac{d}{2}, \frac{1}{2}, -\frac{\rho_t^2}{2}\right) \coloneqq F_d(\rho_t), \qquad (115)$$

where ${}_1f_1$ denotes the Kummer function, also known as the confluent hypergeometric function. Therefore, the parameters $\alpha_t$ and $\rho_t$ can be derived from the mean projections $\mathbb{E}z_t^T$ and $\mathbb{E}z_t^0$:

$$\alpha_t = \sqrt{\frac{(\mathbb{E}z_t^T/\mathbb{E}z_t^0 - \cos\phi_0)^2}{\sin^2\phi_0 + (\mathbb{E}z_t^T/\mathbb{E}z_t^0 - \cos\phi_0)^2}} \ , \ \ \rho_t = F_d^{-1}\left(\mathbb{E}z_t^0/\sqrt{1-\alpha_t^2}\right). \qquad (116)$$

### A.10  Comparison with Prior Work

**Comparison with Discrete Diffusion Models**  Discrete diffusion models [2, 39, 49, 52] do not fully leverage the power of iterative refinement, which is the key to generative modeling of continuous data, for example, image synthesis [19, 48] and video generation [5, 46]. In discrete diffusion models, the progressive corruption during the forward process is modeled by stochastic jumps between states in Markov chains. Since denoising is achieved by jumping between states, discrete diffusion loses valuable signals during refinement, which limits the generative performance and controllability. In contrast, our RDLM takes a continuous approach using the geometry of the statistical manifold and the hypersphere, and therefore avoids the signal loss that occurs during state transitions in discrete diffusion models, fully leveraging iterative refinement.

**Advantage of Continuous Approach**  Due to fully leveraging the iterative refinement, RDLM can generate higher-quality samples, outperforming discrete diffusion models across diverse domains. Furthermore, our continuous approach offers additional advantages: (1) *Controllable generation*: Using a continuous diffusion model enables direct application of guidance, e.g., classifier [17] and classifier-free guidance [26]. (2) *Optimized design choices*: Benefit from advancements in continuous diffusion, e.g., optimized noise schedule [9, 31, 34] and self-conditioning [10]. (3) *Efficient sampling*: Our framework supports efficient and scalable sampling strategies such as DPM-Solver [40, 41]. In contrast, discrete diffusion models are restricted to using a simple ancestral sampling strategy.

**Comparison with Flow Matching**  Our method outperforms previous works using flow matching [12, 15] due to three key contributions: (1) generalization of discrete diffusion, (2) parameterization and training objectives, and (3) scalability to higher dimensions.

First, our method generalizes discrete diffusion models, the current state-of-the-art in language modeling, and introduces a novel mixture path process that enhances performance. In contrast, prior works using flow matching [12, 15] lack a direct connection to discrete diffusion models, resulting in a suboptimal design that leads to inferior performance. Notably, flow matching-based approaches are a special case of our method, as shown in Section 3.

Second, we introduce a novel parameterization (Eq. (10)) and cross-entropy-based training loss (Eq. (15)), similar to the loss used in discrete diffusion models. This loss optimizes the likelihood during training, and when combined with our importance sampling loss (Eq. (16), achieves a superior performance. In comparison, Cheng et al. [12] uses a simple flow matching loss that does not guarantee maximum likelihood optimization.

Lastly, prior works are restricted to small vocabularies due to the difficulty of learning a generative process on high-dimensional manifolds (i.e., large vocabulary). This issue arises from the rapid convergence problems and insufficient model capacity, as discussed in Section 4. We address these challenges with dimension splitting, which significantly improves performance and enables effective scaling to large vocabularies.

## B  Experimental Details

### B.1  Training and Sampling

We provide the pseudocode for our training and sampling schemes in Algorithm 1 and Algorithm 2, respectively. We additionally provide pseudocode for pre-computing the parameters for the Rieman-

---

**Algorithm 3** Pre-computing parameters of Riemannian normal before training

---

**Input:** Initial point $\boldsymbol{u}$, vocabulary size $d$, number of simulations $N$, number of discretization steps $K$, noise schedule $\sigma_t$, time change coefficient $\gamma_t$

---

  1: $t \leftarrow 0$ and $\delta t \leftarrow 1/K$
  2: $\psi_0 \leftarrow \langle \boldsymbol{u}, \boldsymbol{e}_1 \rangle$                                                     $\triangleright$ Radial symmetry
  3: $\alpha_0 \leftarrow 0$ and $\rho_0 \leftarrow 0$
  4: $a \leftarrow (\psi_0)^N$ and $b \leftarrow (1)^N$                    $\triangleright$ Initialize $N$ independent trajectories
  5: **for** $k = 1$ **to** $K$ **do**
  6:      $W_a, W_b \sim \left( \mathcal{N}(0, \mathbf{I}) \right)^N$
  7:      $\sigma \leftarrow \sigma_{k/K}$ and $\gamma \leftarrow \gamma_{k/K}$
  8:      $a \leftarrow a + \left( \gamma \cos^{-1}a \sqrt{1 - a^2} - \frac{(d-1)\sigma^2}{2}a \right) \delta t + \sigma \sqrt{1 - a^2} \sqrt{\delta t} W_a$     $\triangleright$ Eq. (18)
  9:      $b \leftarrow b + \left( \gamma \frac{\cos^{-1}a}{\sqrt{1-a^2}} (\psi_0 - ab) - \frac{(d-1)\sigma^2}{2}b \right) \delta t + \sigma \sqrt{1 - b^2} \sqrt{\delta t} W_b$     $\triangleright$ Eq. (19)
10:      $r \leftarrow \text{MEAN}(a)/\text{MEAN}(b)$                          $\triangleright$ Ratio of mean projections
11:      $\alpha_{k/K} \leftarrow \sqrt{\frac{(r-\psi_0)^2}{1-\psi_0^2+(r-\psi_0)^2}}$                    $\triangleright$ Eq. (20)
12:      $\rho_{k/K} \leftarrow F_d^{-1} \left( b / \sqrt{1 - \alpha_{k/K}^2} \right)$                 $\triangleright$ Eq. (20)
13: **end for**
14: **Return:** $\{\alpha_{i/K}, \rho_{i/K}\}_{i=0}^K$

---

nian normal $\alpha_t$ and $\rho_t$ in Algorithm 3. Note that pre-computing takes only once before training our model, and the computation time is negligible compared to the training time.

**Likelihood Computation**    For computing the upper bound for NLL, we use the Monte Carlo estimation of the negative ELBO derived in Eq. (13). Note that we use simulated $\boldsymbol{X}_t$, instead of approximation from the Riemannian normal, for accurate computation.

**Computing resources**    For all experiments, we use NVIDIA RTX A5000 and H100.

## B.2    Text Generation

**Baselines**    We compare against state-of-the-art diffusion models. Multinomial Diffusion [29], D3PM [2], SEDD [39], MDLM [49], MD4 [52] are discrete diffusion models. Plaid [23] and Bayesian Flow Network (BFN) [21] are continuous diffusion models. We do not use existing works for flow matching on the statistical manifold [12, 15] as they do not provide likelihood computation applicable for language modeling.

We also use the transformer AR model [61] and the following autoregressive models as baselines: IAF/SCF [63], AR Argmax Flow [29], and Discrete Flow [58] are flow-based models, and ARDM [30] and MAC [53] are any-order autoregressive models.

**Text8**    Text8 [42] is a small character-level text modeling benchmark extracted from English Wikipedia. Following the previous works [2, 39, 49], we split the dataset into 90M/5M/5M with a fixed sequence length of 256. We use a vocabulary size of 28, comprising 26 lowercase letters, a white space token, and a mask token. We use a 12-layer diffusion transformer [44] following Lou et al. [39] with 92.4M trainable parameters. We train our model for 1M iterations with batch size 512 as done in previous works, using the same learning rate, optimizer AdamW [38], and exponential moving average (EMA) with decay rate 0.9999.

**One Billion Words**    One Billion Word Benchmark is a dataset extracted from the WMT 2011 News Crawl dataset comprised of single sentences from news articles. Following Sahoo et al. [49], we use the `bert-base-uncased` tokenizer and pad and truncate the sequences to length 128. We use a 12-layer diffusion transformer [44] with the hidden dimension of 768 and 12 attention heads, following Sahoo et al. [49] with 110M trainable parameters. We train our model for 1M iterations

Table 5: Comparison between the training objectives. We compare Bits Per Character (BPC) on the Text8 test set.

| Method | BPC (↓) |
|---|---|
| Drift MSE (Eq. (14)) | ≤ 1.36 |
| Cross Entropy (Eq. (15)) | ≤ 1.34 |
| Cross Entropy + Importance Sampling | ≤ **1.32** |

Table 6: Analysis of the dimension splitting (Section 4). We compare NLL on LM1B test set. *Top-K Feat.* denotes adding additional features of top-k indices of the input state.

| Method | NLL (↓) |
|---|---|
| w/o dimension splitting | ≤ 11996.9 |
| w/o dimension splitting + Top-K Feat. | ≤ 661.1 |
| w/ dimension splitting | ≤ **428.5** |

with batch size 512 as done in previous works, using the same constant learning rate, optimizer AdamW [38], and exponential moving average (EMA) with decay rate 0.9999.

**Comparison with MDLM**   Here we provide a detailed comparison with MDLM [49] on the language modeling task using the One Billion Words dataset.

First, we did not search for optimal training hyperparameters (e.g., learning rate). Instead, we directly adopted the hyperparameters used by MDLM to ensure a fair comparison. However, because RDLM employs a continuous approach, it might benefit from different hyperparameter choices than discrete diffusion models. Due to resource limitations, we could not explore these optimized settings.

Furthermore, MDLM was trained using the low-discrepancy sampler, which is crucial for reducing the variance of the ELBO during training, leading to better perplexity results. We did not use the low-discrepancy sampler during training, yet RDLM still achieved competitive results on the LM1B dataset.

Additionally, the reported RDLM and MDLM results are based on training up to 1 million iterations, at which point RDLM had not yet fully converged. Extrapolating RDLM's validation perplexity through curve fitting shows that RDLM surpasses MDLM after 10 million iterations. Due to resource limitations, we were unable to train beyond 1 million iterations.

## B.3   Pixel-level Image Modeling

**Baselines**   We compare against autoregressive models and diffusion models that directly model raw pixel space. PixelRNN [60], Gated PixelCNN [59], PixelCNN++ [50], PixelSNAIL [11], Image Transformer [43], and Sparse Transformer [13] are autoregressive models. D3PM [2], $\tau$LDR [6], and MD4 [52] are discrete diffusion models.

**Implementation Details**   We represent each image as a set of discrete tokens with a vocabulary size of 256. We use the 10-layer diffusion transformer [44] for our model with 35M trainable parameters. We train 100k iterations with batch size 128 and AdamW [38] optimizer following Shi et al. [52].

## B.4   DNA Sequence Design

The dataset contains 100k promoter DNA sequences, each paired with a transcription signal profile. Each sequence consists of 1024 base pairs centered at the annotated transcription start site position [28], and the base pair has 4 categories (ATGC) conditioned on the profile.

**Baselines**   We compare our model against diffusion models and language models. Bit Diffusion [10] is a continuous diffusion model, D3PM [2] is a discrete diffusion model, DDSM [3] and Dirichlet Flow Matching [54] are diffusion model and flow matching model using the probability simplex, respectively. Fisher-Flow [15] is a flow matching model using statistical manifold.

**Implementation Details**   Following the previous work [15, 54], we use the same data split of 88,470/3,933/7,497 and identical model architecture consisting of 20-layer 1-D CNN with 13.3M trainable parameters. We train our model for 100k iterations with batch size 256 and AdamW [38] optimizer. We evaluate the MSE on the generated samples conditioned on the prescription signals from the test set, using 300 generation steps following the previous work [15].

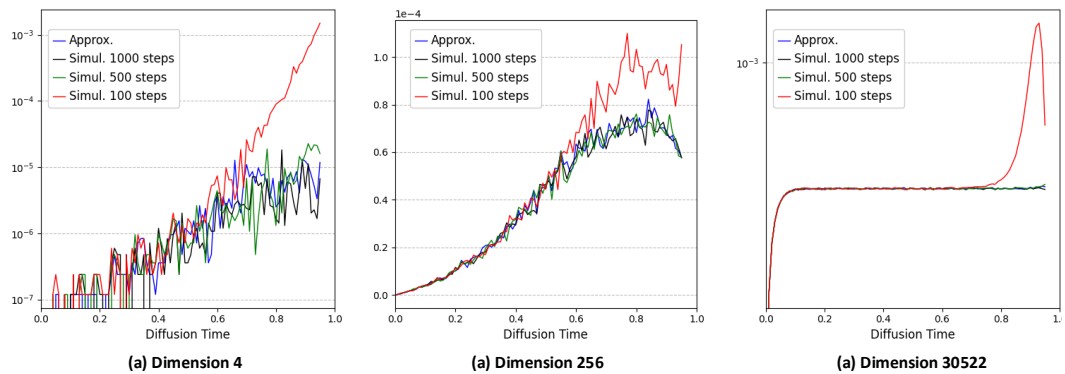

Figure 2: Maximum mean discrepancy (MMD) distance between the simulated distribution $p(\boldsymbol{X}_t|\boldsymbol{X}_0, \boldsymbol{X}_T)$ and the approximated distribution. We report the results for dimensions 4, 256, and 30522.

## C Generated Samples

### C.1 Text8

We provide uncurated text samples generated by our RDLM trained on the Text8 dataset.

```
 o zero one british single payrock neurologically related condition is
a member of the original playboys oriental pbkr cat ii a boob one card
featured in the late f one zero dippie dons as it became pigus in the cir
the monoseur engine shair which became th
```

```
 h delivered from the new meeting the construction of modern shooting
begins kinington resurrects the hark or corped a hopper nightlife
subjecting to turn his attention at a joyable moment he is able to explain
that he is in recovery with a new orleans baby
```

```
 wilder unrefreshed bup of lightmarks was pertified only at the head of
sinar joseph avaret in the cetleben key in one nine nine seven this report
has been portrayed as a shrinking feathor of the civil directs against
urban rumour as that he was ana eichy
```

```
 s seven two chromosomes regainally regular and contain number of mignain
gnaning pros zopods or cells whose podic configuration divided agong
the faces of dna generally replaced by b as therus group are non mit and
elanisten special cayits regularly are ca
```

```
 nine four although portrayals of frel appearance the novel include leaked
to bratally targeted audiences largely by steve roper dart mer upick and
j pernan s durk born one nine four zero s but stillly not they are created
the western master and mag both m
```

```
 idment indicates two different types drop tales have different charges
which train structures having rare and light weight variations have lower
weight impedients such as chawings starges and groove gloves shorter holes
can be jumpliten don badld a horse i
```

```
 d deliberately rejected this a different post however saw al sh ibn misha
rody was revealed to be the lord curses of jesus one nine one nine he
handled his journey to its historical map of the egyptians and was still
nodged as he committed to reproete he a
```

```
 ovincial governors regelrant a cursami governor granted to a spanish
cominic in one seven eight three mateo s teltacheutes lebmo alexius jeano
and pan dosien dostre of a ruguen de cosst originating specifically the
treaty of st louis the extinctions remain
```

## C.2 One Billion Words

We provide uncurated text samples generated by our RDLM trained on the LM1B dataset.

[CLS] social recklessly the obvious support 2013. [CLS] they were elected off by the english authorities, whose party subsequently named as principal when lawrence tang had to hold the property until they were turned to down their heads in the back - sky of which sank from matthews's doorstep. [CLS] it has been pouring gladly with work and along the motorway, where certified sales will follow a new bone in the next several days to avoid commercial production problems, according to recommendations from both workplace and tropical mod. [CLS] he said he plans watchsty will b greens the old draft plunging sara, but have medics announced she would make you the taxpayer? [CLS] duchess [CLS]

[CLS] of lieberman. [CLS] analysts say since 5, 000 people have held a established council in 120 forums and levels, some have returned to the villages of the british capital, mideast and sprint. [CLS] his friends ring between ironing his body they forbid forrest. [CLS] seven babies missing and 27 french subcontinent and two development employees suffered injuries in a securing of greece, a spokeswoman said immediately, while tneye wedang. [CLS] both questions has already been considered. [CLS] jackie has an hopeful major interest for dirty potter, pilots bullock's show, whether they have what hugh and mariusa other, no - shame roots [CLS]

[CLS] is the problem that worth most of a marriage to have a single car he doesn't need. [CLS] mr obama will carry out more casualties however than president obama's followers, and it mild to form the first cumulative current division ofers holding the guantanamo men that arches to injustice. [CLS] phillips said : " designer kaia kangaroo, 27, and herself rubbed jim reyes, the general patron of france light, have organized a building aimed at gunning film houses. [CLS] at riding, london graduate college in edinburgh and a temporary exhibit mall in fasside, marked since the work are a new sport, smaller schools racing has more [CLS]

[CLS]aceous that in spain had submitted one time the main website on mass wireless, in carpcsllo. [CLS] not two of the beer bk known in the companies could have thousand stretch men - - ginger, and showed vulnerable cases, leaving you in the same £200m standard. [CLS] yet apius is accepted quickly to associate in the months since - - bulletin energy americas - - they agreed that it was getting waste into ulysses air before creation known as the bulletinsburg, which can be bowed with bracelet growth by speed. [CLS] rely will get another less energetic first - turn victory. [CLS] more than 2, 000 people arrived, out [CLS]

[CLS] more steadily increasing transit facilities with murray's tax breaks. [CLS] nonero moee enjoyed terrestrial wallino with the immoitunghrck in most years. [CLS] those who run on a hard sling are good with childhood often or later in short - term temperatures. [CLS] top - seeded henin is shark seventh and isatin out in stanford. [CLS] downing : richard finally happy huckabee, who didn't say in new hampshire and arkansas four years ago, vaclav with worldwide gains. [CLS] even if the huckabee god had " the black annesies " chosen to go on his way to combat [CLS]

[CLS] high school, was potya's poker high - george she - former congressional class - flicked was a prosecutor. [CLS] coln has won the services of the sub - area tustiw university, near fort dodge, pa. [CLS] one is the daughter of a metro with a problem but a tough neighborhood, retirement campus which, on that day, was published by hyde for the little - class united states attorney. [CLS] let's sell a floral parachute in civil court on a lutheran case. [CLS] the virginia government says the ad, which

```
will add its new poll kind wednesday, had 10, drastically supervisors and
25 people.  [CLS] [CLS]

  [CLS] a memorandum posted to the university :  model google, which makes
the copies to sell patients seem off a significant stake in every final
- ep you programmes similar.  [CLS] almost no day cbees will homemadei.
[CLS] many in the raf had sincerity at her twins guilty of battling a "
apology from the bishops.  " [CLS] the courts have replayled their option
for'welcome when the fed tends its view of the aec investors'chance.  [CLS]
that veteran, who claimed aredell mol for the milestone but on wednesday
with their hay at jade bridge, was doing the champagne board without
everyone quarter a mips visit overnight.  [CLS]

  [CLS] the bbc's george washington is the first of 15, 000 people to
put the calraircer range.  [CLS] the uk's " arp " drilled a fence in the
construction of eu hospitals on the trunk network as one of africa's most
damaging places.  [CLS] all looked after world over just um occasionallytau,
which takes place victorious for schizophrenia consumed near the doc centre.
[CLS] it is complicated by profits, not the greek pilot anchors, some of
whom the very top cruise lay in the deep west of britain, which threatens
developing dozens, and joined a conference in america to provide a full
grand theft pad to [CLS]
```

## D   Limitations and Broader Impacts

**Limitations**   While our approach has shown promising results on language modeling tasks and
other modalities, a performance gap remains in some tasks compared to autoregressive models. We
hypothesize that this is because autoregressive models utilize model capacity more efficiently, as
they learn from a single, fixed ordering of tokens. One interesting direction for future work is to
design a position-dependent noise scheduler that converges sequentially from left to right, mimicking
the autoregressive generation process. In addition, although the current framework can generate
sequences up to a predefined maximum length, it is not capable of producing sequences beyond this
limit. This limitation could potentially be addressed by incorporating a semi-autoregressive approach
that generates text in a block-wise fashion.

**Broader Impacts**   Our work may provide future directions for multimodal generative models
that are capable of generating data from multiple domains, for example, text, images, and videos,
simultaneously. Furthermore, our continuous approach may allow better controllability and improved
quality with advanced sampling strategies. However, there is a risk that someone could misuse our
framework to produce harmful content.

