# OpenReview forum: "Continuous Diffusion Model for Language Modeling"
_NeurIPS.cc/2025/Conference — NeurIPS 2025 poster_

### Official Review · Reviewer_FzDR · 2025-06-30

**Clarity:** 2
**Significance:** 2
**Originality:** 3
**Rating:** 3
**Confidence:** 2

**Summary:**

This article proposes a continuous diffusion model for language modeling, which integrates the geometry of the underlying categorical distribution. It establishes a link between discrete diffusion and continuous flow on the statistical manifold, and based on this connection, introduces a simple diffusion process that generalizes existing discrete diffusion models. The article also puts forward a simulation-free training framework relying on radial symmetry, along with a straightforward technique to tackle the high dimensionality of the manifold. Experiments across language modeling benchmarks and DNA are conducted, showing that the proposed method surpasses existing discrete diffusion models.

**Questions:**

see weaknesses

**Ethical Concerns:**

["NO or VERY MINOR ethics concerns only"]

**Final Justification:**

My point of view has not changed; I still value the novelty of this paper and question its practical usage. My score is pretty marginal, and I keep my score between 3 and 4.

**Limitations:**

see weaknesses

**Quality:**

2

**Strengths And Weaknesses:**

Strengths:
- The methodology of connecting discrete simplex modeling to the generative modeling on the sphere manifold is novel and reasonable, at least for me.
- The mathematical derivation is solid, though I didn't check the detailed proofs.

Weaknesses:
- I am not familiar with the language tasks, but I think I witnessed some more large-scale language generation benchmarks in the technical report of Qwen3 and Kimi-chat. I think the method in this paper can not scale to that level, and it is a little overclaim to say 'approaches the performance of autoregressive models.'.
- Detailed time-cost of RDLM compared to other discrete diffusion models and DDPM is needed.
- On CIFAR-10 experiments, the comparison to classical DDPM/EDM/rectified flow matching is beneficial.
- Experiments on scalability should be given.

---

> ### Author Rebuttal · Authors · 2025-07-29
>
> We sincerely thank you for your constructive and helpful comments. We appreciate your positive feedback that our work is novel and reasonable, and the mathematical derivation is solid. We initially address your comments below:
>
> ---
> >**C1:** I think the method in this paper can not scale to Qwen3 and Kimi-chat, and it is a little overclaim to say 'approaches the performance of autoregressive models.'
>
> **R1:** Thank you for your valuable feedback. We would like to clarify that our work proposes a novel language diffusion framework aimed at improving previous diffusion models for language modeling. Our primary focus is on advancing the modeling approach rather than scaling to very large modern LLMs such as Qwen3 or Kimi-chat.
>
> Regarding the comparison to autoregressive models, this comparison is made within the context of models of similar scale. We do not claim that our method matches the performance of state-of-the-art large-scale autoregressive models, but rather that it approaches the performance of autoregressive models at comparable model sizes. We will clarify this point in the revised manuscript to avoid any overclaim.
>
> ---
> >**C2:** Detailed time-cost of RDLM compared to other discrete diffusion models and DDPM is needed.
>
> **R2:** RDLM does not require additional computational cost during sampling compared to the discrete diffusion models in terms of the number of model evaluations. Specifically, to generate samples, RDLM simulates the generative process modeled by the SDE in Eq. (6) (please see Algorithm 3 in the Appendix), and simulating Eq. (6) with a fixed step solver (e.g., Euler-Maruyama) requires the same number of model evaluations as simulating the discrete diffusion process. Therefore, in terms of sampling time-cost, RDLM is comparable to discrete diffusion models, as both approaches rely on an equivalent number of model evaluations per sample generation.
>
> ---
> >**C3:** In CIFAR-10 experiments, the comparison to classical DDPM/EDM/rectified flow matching is beneficial.
>
> **R3:** CIFAR-10 experiment in our work addresses a fundamentally different task compared to the image generation tasks tackled by DDPM or EDM models. Our CIFAR-10 experiment focuses on pixel-level image modeling framed as an order-agnostic set modeling problem of discrete tokens, each drawn from a vocabulary of size 256, following the approach in MD4 (Shi et al., 2024). This formulation intentionally removes spatial information about pixel proximity, which is a key inductive bias exploited by classical image diffusion models through architectures like U-Net. Given this distinct setting, we excluded DDPM and EDM models from our baselines, as their assumptions and modeling strategies do not directly apply.
>
> ---
> >**C4:** Experiments on scalability should be given.
>
> **R4:** Our experiment on the LM1B dataset, which features a sufficiently large vocabulary of size 30,522, demonstrates strong potential for scaling to even larger vocabularies. Notably, RDLM is designed to scale to higher dimensions efficiently due to its simulation-free training scheme (as detailed in Section 4) and the dimension splitting technique (described in Section 5).

---

> ### Comment · Reviewer_FzDR · 2025-08-06
>
> Thanks for your detailed rebuttal.
>
> My point of view has not changed; I still value the novelty of this paper and question its practical usage.
> My score is pretty marginal, and I keep my score between 3 and 4.

---

> > ### Author Response · Authors · 2025-08-06
> > **Thank you for recognizing the novelty of our work**
> >
> > We sincerely thank you for recognizing the novelty of our work. We appreciate your thoughtful feedback and understand your concerns regarding its practical usage.
> >
> > We respectfully ask that, as you reflect on your final assessment, you kindly take into account:
> > - Novelty and technical contribution of our work, and
> > - Strong potential for practical impact as evidenced in the experiments on LM1B dataset.
> >
> > Thank you again for your time and thoughtful review.

---

### Official Review · Reviewer_iVwZ · 2025-07-02

**Clarity:** 2
**Significance:** 2
**Originality:** 2
**Rating:** 2
**Confidence:** 5

**Summary:**

The paper proposes Riemannian Diffusion Language Model (RDLM), a continuous diffusion model that treats each categorical token as a point on the positive orthant of a hypersphere.

Key contributions of the work include:

1. proving an explicit correspondence between discrete masked / uniform diffusion and geodesic flows on the statistical manifold
2. designing a mixture path that smoothly blends the two processes to enable gradual refinement
3. deriving a simulation‑free training objective that leverages radial symmetry to sample bridge states cheaply
4. using dimension‑splitting that maps large vocabularies to products of low‑dimensional spheres for stable learning

Experiments on Text8, LM1B, CIFAR‑10, and promoter‑DNA design show that RDLM is competitive with strong discrete diffusion baselines.

**Questions:**

1. What happens if you use only the masked or only the uniform path? Include an ablation on LM1B; a clear gain would strengthen originality claims.
2. Please clarify theoretical differences with GIDD and give a quantitative comparison.
3. Did you run standard Gaussian diffusion on the simplex or the sphere before implementing the Riemannian diffusion approach?

**Ethical Concerns:**

["NO or VERY MINOR ethics concerns only"]

**Final Justification:**

I appreciate the authors' response, but the core issue of baseline comparison remains unresolved. A key motivation for this work is the claim that "Existing continuous diffusion models for discrete data underperform compared to discrete methods." However, this foundational assertion is not substantiated with references to the literature, nor is it validated empirically through direct comparisons with leading continuous methods like LD4LG and TEncDM.
Without such an evaluation, the paper fails to convincingly demonstrate superiority over other continuous diffusion methods.

**Limitations:**

The dense theoretical exposition currently obscures the core ideas of the model. I recommend moving some of the derivations to the appendix and using the freed space to present more experimental analysis and comprehensive ablation studies in the main text.

Please refer to the questions and weaknesses outlined above for further details.

**Quality:**

3

**Strengths And Weaknesses:**

**Strengths**

1. By grounding diffusion in information geometry, the work unifies discrete and continuous views and offers a principled route to exploit iterative refinement without losing signal at state jumps.
2. The paper contains detailed derivations of the flow equivalence, likelihood bound, and bridge approximation, with formal proofs moved to the appendix .
3. Results on images and DNA sequences suggest the method works across different kinds of data.


**Weaknesses**

1. Important continuous text diffusion baselines such as [DiffuSeq](https://arxiv.org/abs/2210.08933), [TEncDM](https://arxiv.org/abs/2402.19097), and [LD4LG](https://arxiv.org/abs/2212.09462) are missing, making it hard to gauge relative progress.
2. [GIDD](https://arxiv.org/abs/2503.04482), that also suggests combining masked and uniform discrete diffusion, is not discussed.
3. The language results are limited to Text8 and LM1B, which are both relatively small and dated corpora. Other commonly used benchmarks for text diffusion models, such as RocStories, and OpenWebText, are not covered. This omission makes it difficult to compare the proposed method with other diffusion-based baselines. Furthermore, the field has advanced considerably, and evaluation on more challenging tasks, including conditional generation, would be necessary to demonstrate the method’s practical relevance.
4. The work claims that the continuous approach offers advantages for controllable generation but provides no evidence on conditional text generation tasks, such as summarization.
5. The bridge approximation is empirically validated with MMD, yet its impact on final likelihood in extreme dimensions is not thoroughly analysed.
6. The paper also lacks a comparison with Gaussian diffusion on the simplex as an alternative approach to modeling discrete data, as explored in methods such as [SSD-LM](https://arxiv.org/abs/2210.17432) and [TESS](https://arxiv.org/abs/2305.08379).

---

> ### Author Rebuttal · Authors · 2025-07-29
>
> We sincerely thank you for your constructive and helpful comments. We appreciate your acknowledgement of our key contributions, in particular, that our work offers a principled route to exploit iterative refinement without losing signal at state jumps. We initially address your comments below:
>
> ---
> >**C1:** Missing related works
>
> **R1:**
> We appreciate the reviewer for providing additional related works on text generation. We would like to clarify that DiffuSeq is a diffusion model designed for sequence-to-sequence tasks, whereas our work addresses general unconditional text generation. Moreover, TEncDM and LD4LG are continuous diffusion models that adapt image diffusion models to text, and they are conceptually similar to Diffusion-LM, Plaid, or Bayesian Flow Network, which we have included as baselines in our comparisons. We will add discussion with these works in the revision.
>
> ---
> >**C2:** Difference with GIDD
>
> **R2:** We appreciate the reviewer for bringing to our attention the very recent work GIDD, which is a contemporaneous work (appeared on arXiv after March 1st). GIDD is a discrete diffusion model that interpolates masked and uniform diffusion by mixing masking and uniform noise. On the other hand, our RDLM is a continuous diffusion model using Riemannian diffusion process on hypersphere, which fully exploits the power of iterative refinement.
>
> In particular, our mixture path formulation (Eq. (8)) can be seen as a continuous generalization of interpolating diffusion process in GIDD. Extending our Proposition 3.1., the mixture path built from mixing masked diffusion and uniform diffusion corresponds to the trajectory of transition distribution of GIDD process. Moreover, Eq. (8) allows interpolating more than two processes, while GIDD is limited to mixing masked and uniform noise.
>
> During the rebuttal, we could not make a quantitative comparison with GIDD as GIDD does not have checkpoints for the benchmarks used in our work, for example, Text8, LM1B, CIFAR-10, or DNA sequence design. We will include a detailed discussion with GIDD and add comparison in the final revision.
>
> ---
> >**C3:** Language results are limited to Text8 and LM1B and other benchmarks such as RocStories and OpenWebText, are not covered
>
> **R3:** Text8 and LM1B are among the most widely used datasets for evaluating diffusion models in recent literature, especially for discrete diffusion models. Notable recent works such as SEDD (Lou et al., 2024), MDLM (Sahoo et al., 2024), and MD4 (Shi et al., 2024) also use these datasets for benchmarking. These works do not typically use RocStories as a benchmark. We believe that experiments on Text8 and LM1B, as well as image generation and DNA sequence design extensively validate the effectiveness of our framework. While we agree that results on OpenWebText would further strengthen our work, we could not run the experiments due to limited resources. We will make every effort to include OpenWebText results in the final revision.
>
> ---
> >**C4:** Evaluation on more challenging tasks, including conditional generation, would be necessary
>
> **R4:** Unconditional generation of discrete data using diffusion models is itself a challenging task, with a notable performance gap compared to autoregressive models. Recent works such as MDLM, MD4, and Discrete Flow Matching, have also focused primarily on unconditional generation tasks without extending to conditional generation. In our paper, we focus on validating the effectiveness of our new continuous diffusion framework for modeling language and discrete data in an unconditional setting. While conditional generation tasks are important, addressing them is beyond the scope of the current work.
>
> ---
> >**C5:** The work claims that the continuous approach offers advantages for controllable generation but provides no evidence on conditional text generation tasks
>
> **R5:** We would like to clarify that we did not claim our method to have advantage for controllable generation. In introduction lines 29–30, we explained that previous works have adapted continuous diffusion models for discrete data by their advantage on controllability. For example, Diffusion-LM uses classifier-guidance to diffusion model for controllable text generation. While our continuous diffusion process can similarly incorporate classifier-guidance, we have not explored this direction in the current work.
>
> ---
> >**C6:** The bridge approximation is empirically validated with MMD, yet its impact on final likelihood in extreme dimensions is not thoroughly analysed.
>
> **R6:** MMD results show that our approximation on the transition distribution is accurate. This leads to superior results on test perplexity in the LM1B dataset, which has a high dimension of 30522.
>
> ---
> >**C7:** Comparison with Gaussian diffusion on simplex or sphere
>
> **R7:**
> First, we would like to clarify that Gaussian diffusion are ill-defined on non-Euclidean space like simplex or sphere. This is because the Gaussian distribution itself is not well-defined in these spaces, and diffusion process must be redefined to account for the underlying geometry and metric of the manifold.
>
> While Gaussian diffusion models can be applied in the ambient Euclidean space, recent literature on generative modeling over manifolds, for instance, RDM (Huang et al., 2022) or RSGM (De Bortoli et al., 2022), demonstrate that Riemannian diffusion outperforms Gaussian diffusion when modeling data that reside on non-Euclidean manifolds. Our experimental results further support this, showing that our method significantly outperforms Diffusion-LM, a Gaussian diffusion model designed for text data.
>
> Furthermore, prior works used as baselines in our paper, for example, DirichletFM (Stark et al., 2024) and Fisher-Flow (Davis et al., 2024), show that naively adapting flow matching to simplex, i.e.., LinearFM, underperformed Riemannian approaches. We will include this discussion in our revision and add LinearFM as our baseline.
>
> ---
> >**C8:** What happens if you use only the masked or only the uniform path?
>
> **R8:** When using only the masked or the uniform path, we achieved test perplexity of 1.40 and 1.41 on the Text8 test set, respectively, while using the mixture path achieved 1.32.

---

> > ### Comment · Reviewer_iVwZ · 2025-08-01
> > **I thank the authors for their rebuttal. While I appreciate the clarifications, some important points were not sufficiently resolved. I will reiterate my concerns below.**
> >
> > > **C1:** Missing **baselines**
> >
> > I must respectfully disagree with the authors' characterization of the cited papers. My initial point was that LD4LG and TEncDM are not merely "related works" to be discussed, but crucial baselines for any new continuous diffusion model for text. A discussion is insufficient; a direct empirical comparison is required for a rigorous evaluation.
> >
> > Additionally, I would like to highlight an inconsistency in the current argument. The paper's core motivation, stated in the abstract, is that “Existing continuous diffusion models for discrete data underperform compared to discrete methods.” However, to validate this claim, the work must be benchmarked against the strongest continuous models available. By omitting direct empirical comparisons with state-of-the-art methods like LD4LG and TEncDM, the paper fails to support its own central premise.
> >
> > Furthermore, the claim that these models are "conceptually similar" to existing baselines is a significant oversimplification. If similarity is defined by the use of Gaussian diffusion, the term becomes too broad to be meaningful. TEncDM's contribution, for instance, was in demonstrating that diffusion in the context-aware encoding space surpasses earlier methods. Similarly, LD4LG introduced a latent diffusion approach that proved highly effective. Both methods reported substantial improvements over older models like Diffusion-LM, which makes relying on Diffusion-LM as a primary point of comparison insufficient.
> >
> > Finally, framing these works as simply "adapting image diffusion models to text" misrepresents their specific contributions to natural language generation and is not a valid reason for their exclusion as baselines.
> >
> > > **C3:** Language results are limited to Text8 and LM1B and other benchmarks such as RocStories and OpenWebText, are not covered
> >
> > I acknowledge the authors' point that Text8 and LM1B are common benchmarks in recent literature. However, the cited examples are all discrete diffusion models. The critical baselines for this work are other *continuous* diffusion models, such as Diffusion-LM and the ones I previously mentioned (LD4LG, TEncDM), which have established RocStories as a standard benchmark.
> >
> > Furthermore, the absence of results on a large-scale corpus like OpenWebText raises significant questions about the practical applicability and scalability of the proposed method. I appreciate the authors' transparency about resource constraints and their commitment to including these experiments in the future. Nevertheless, without these results, the scalability and real-world performance of the model remain open questions.
> >
> > > **C7:** Comparison with Gaussian diffusion on simplex or sphere
> >
> > I acknowledge the theoretical point that Gaussian diffusion is ill-defined on non-Euclidean manifolds like the simplex. However, this theoretical limitation does not justify the exclusion of empirically successful baselines. Recent works, namely SD-LM (Han et al., 2022) and TESS (Mahabadi et al., 2023), have demonstrated that applying Gaussian diffusion on simplex is a competitive strategy for text generation.

---

> > > ### Author Response · Authors · 2025-08-01
> > > **Thank you for your prompt response.**
> > >
> > > We sincerely thank you for your prompt response.
> > >
> > >  ---
> > > >**C9:** Comparison with LD4LG or TEncDM
> > >
> > > **R9:** We appreciate the valuable feedback. We would like to clarify in detail why we did not use LD4LG and TEncDM as baseline in our experiments.
> > >
> > > LD4LG and TEncDM both rely heavily on large pre-trained language models as their encoder and decoder. LD4LG utilizes BART or T5, and TEncDM uses BERT. In contrast, our approach does not leverage such pre-trained language models.
> > >
> > > This distinction is crucial, as pre-trained language models are trained on vast external corpora, providing them with significant prior knowledge that are not present in the limited dataset used to train our model and the baselines.
> > >
> > > As a result, direct comparisons would not be fair or informative, as the performance of LD4LG and TEncDM is heavily influenced by the knowledge encoded in their pre-trained components, rather than solely by the diffusion modeling approach.
> > >
> > > For a fair and meaningful evaluation, we believe it is important to compare models under similar training data and architectural assumptions. This is why we chose Diffusion-LM, Plaid, BFN as baselines, as they are more directly comparable to our approach in the context of continuous diffusion models.
> > >
> > >
> > > ---
> > > >**C10:** Without results on large-scale corpus like OpenWebText, the scalability and real-world performance of the model remain open questions.
> > >
> > > **R10:** Thank you for your feedback regarding evaluation on large-scale corpora. In our work, we validated our approach on the LM1B dataset, which is a widely recognized real-world dataset that features a sufficiently large vocabulary and a substantial number of training examples, making it a strong proxy for large-scale language modeling tasks. Our results on LM1B demonstrate that our method scales effectively to large datasets and realistic settings. While we agree that additional experiments on even larger corpora such as OpenWebText could further strengthen our work, we believe that comprehensive evaluation on LM1B provides strong evidence of the scalability and practical applicability of our model.
> > > We hope this addresses your concern and clarifies the real-world relevance and scalability of our approach.
> > >
> > > ---
> > > >**C11:** Comparison with simplex diffusion models
> > >
> > > **R11:** Thank you for highlighting additional related works. We excluded these works from our main comparison because their generation styles and objectives differ from ours:
> > >
> > > SSD-LM is a semi-autoregressive diffusion model, whereas our RDLM and the diffusion language models used as baselines are non-autoregressive. Recent literature, such as SEDD (Lou et al., 2024) and MDLM (Sahoo et al., 2024), demonstrates that discrete diffusion models outperform SSD-LM. Since our method matches or surpasses these state-of-the-art models, it is reasonable to conclude that our approach also outperforms SSD-LM.
> > >
> > > TESS is a sequence-to-sequence diffusion model designed for sequence-to-sequence generation tasks, while our work focuses on general unconditional text generation. This fundamental difference in task formulation led to excluding TESS from the baselines.
> > >
> > > ---
> > >
> > > We hope our response addressed your concerns. We again thank you for your constructive feedback.

---

> > > > ### Comment · Reviewer_iVwZ · 2025-08-05
> > > >
> > > > I appreciate the authors' response, but the core issue of baseline comparison remains unresolved. A key motivation for this work is the assertion that "Existing continuous diffusion models for discrete data underperform compared to discrete methods," yet this claim is not supported by direct comparison with leading continuous methods like LD4LG and TEncDM.
> > > > The authors' rationale for this omission—the use of pre-trained backbones in those models—is unpersuasive. These components are openly accessible, and their use is standard for achieving better performance.
> > > > Without such an evaluation, the paper fails to convincingly demonstrate superiority over other continuous diffusion methods.

---

> > > > > ### Author Response · Authors · 2025-08-06
> > > > > **We appreciate the reviewer's thoughtful feedback**
> > > > >
> > > > > We appreciate the reviewer's thoughtful feedback. However, we respectfully disagree regarding the necessity of a direct comparison with LD4LG and TEncDM. These models significantly benefit from the prior knowledge of pre-trained language models, which introduces a confounding factor when assessing the core contribution of diffusion modeling techniques.
> > > > >
> > > > > To accurately evaluate the effect of the diffusion modeling approach itself, it is important to conduct controlled comparisons among models trained under the same data constraints without the advantages conferred by large-scale pre-training, which is the case for Diffusion-LM, Plaid, and Bayesian Flow Network.
> > > > >
> > > > > We sincerely hope the reviewer will consider this point in the final assessment. Thank you once again for your valuable feedback.

---

### Official Review · Reviewer_myHk · 2025-07-02

**Clarity:** 2
**Significance:** 3
**Originality:** 3
**Rating:** 5
**Confidence:** 3

**Summary:**

This work studies the continuous diffusion modeling for the discrete language task and presents the Riemannian Diffusion Language Model (RDLM) based on solid theoretical guarantees. More specifically, they first show the relationship between continuous flow on the manifold. Based on this, this paper develops a complete training framework to train RDLM and achieve excellent results in text, image, and DNA sequence tasks.

**Questions:**

1.	In line 726 of this paper, this work says that the mixture path process enhances the performance. Is this augmentation supported by experimental evidence (for example, RDLM without mixture path)?

2.	As this work shows the potential of RDLM, would it be possible to scale up RDLM to achieve great performance?

**Ethical Concerns:**

["NO or VERY MINOR ethics concerns only"]

**Final Justification:**

As showing in the discussion phase, I still think the modeling for the continuous diffusion models for the text generation task. Hence, I vote to accpet this work. However, as discussed by Reviewer iVwZ, the extension to the latent space is also important and interesting in leading to a large scale continuous diffusion models for text tasks.

**Limitations:**

yes

**Quality:**

4

**Strengths And Weaknesses:**

Strength:

1.	Each component of RDLM (including the theoretical foundation, mixture paths, and training framework) has been discussed and designed in detail.

I didn't find obvious weaknesses; I just have some questions (in the next section).

A minor Weakness:

1.	It would be better to add more visualizations for an easier understanding of the RDLM pipeline.

---

> ### Author Rebuttal · Authors · 2025-07-29
>
> We sincerely thank you for your constructive and helpful comments. We appreciate your positive comments that our work is based on solid theoretical guarantees, achieve excellent results, and each component is discussed and designed in detail. We initially address your questions below.
>
> ---
> >**C1:** It would be better to add more visualizations for an easier understanding of the RDLM pipeline.
>
> **R1:** Thank you for your valuable feedback. Due to the page limit, we were unable to include additional visualizations in the current version. We agree that visual aids would enhance understanding, and we will add concept figures illustrating the RDLM pipeline in the final revision.
>
> ---
> >**C2:** Comparison of mixture path and masked or uniform path
>
> **R2:** When using only the masked or the uniform path, we achieved test perplexity of 1.40 and 1.41 on the Text8 test set, respectively, while using the mixture path achieved 1.32. This shows that using the mixture path results in significantly improved performance.
>
> ---
> >**C3:** As this work shows the potential of RDLM, would it be possible to scale up RDLM to achieve great performance?
>
> **R3:** Indeed, RDLM has the potential to achieve better performance when scaled up with larger models. Larger models can more accurately learn the transition distribution $p_{T|t}(X_T|X_t)$, which is critical for the effectiveness of RDLM. Our empirical results on Text8 and CIFAR-10 datasets support this observation, where we observe improved performance with increased model size. We believe that further scaling of RDLM could lead to even greater performance gains.

---

> > ### Comment · Reviewer_myHk · 2025-08-06
> >
> > Thanks for the author's response. After reading the rebuttal and other reviewers' discussion, I have the following comments. When considering the full space diffusion models in text generation, it is fair to compare with Diffusion-LM, and the modeling of RDLM is interesting. However, as we know, learning in the latent space is a standard method in computer vision generation. Hence, it is also interesting to elucidate whether RDLM can be done or comparable with LD4LG and TEncDM (since I think this is a promising way to scale up the dLLMs).

---

> > > ### Author Response · Authors · 2025-08-06
> > > **Thank you for your constructive comments**
> > >
> > > We sincerely thank you for your constructive comments.
> > >
> > > We appreciate your acknowledgement that our experimental setup of comparing with Diffusion-LM is fair. A direct comparison with LD4LG or TEncDM, which use pre-trained language models, is not fair as these methods benefit from extensive prior knowledge acquired from large external corpora. In contrast, our model and the baselines are trained solely on the limited dataset, without access to such external resources.
> > >
> > > We agree with your suggestion that extending our RDLM to representation space generated by pre-trained language model will be a promising direction for scaling up diffusion language model. We would like to explore this direction in future work.

---

> > > > ### Comment · Reviewer_myHk · 2025-08-07
> > > >
> > > > I still maintain my score to accept this work as I think the modeling and framework are interesting enough, even in the full space instead of the latent space (from the intuition, I also think this modeling can easily extend to the latent space). Good luck.

---

### Official Review · Reviewer_GUHT · 2025-07-03

**Clarity:** 3
**Significance:** 3
**Originality:** 3
**Rating:** 4
**Confidence:** 3

**Summary:**

This paper proposes a continuous diffusion model for language modeling by projecting the categorical distribution to the positive orthant of a sphere, which is diffeomorphic to the manifold of distributions. This enables the benefit of continuous diffusion, such as efficient sampling. To make it practical for language modeling, several tricks are introduced, such as dimension splitting. Experiments on text and image demonstrate the potential of the proposed method.

**Questions:**

- Line 135: It says "providing numerous opportunities to correct wrong predictions during the process." I am unsure about this claim, because being continuous can also lead to error accumulation, while discrete transitions hold the possibility of entirely denoising the accumulated error.
- Eq. (10): The left-hand side should be $p\_\\theta(X\_T)$? Since this is a distribution over $X\_T$, it would be better to use a clearer notation for this.

**Ethical Concerns:**

["NO or VERY MINOR ethics concerns only"]

**Final Justification:**

The authors addressed some of my concerns, therefore, I intend to maintain my original positive rating.

**Limitations:**

yes

**Paper Formatting Concerns:**

Format looks good to me.

**Quality:**

3

**Strengths And Weaknesses:**

Strengths:
- The paper tackles continuous diffusion language models, which is an important topic in current AI research.

Weaknesses:
- While the proposed method is elegant and neat, it is unclear whether there is real benefit for large-scale language modeling. The experiments on language modeling remain relatively small-scale.
- The concern about its effectiveness in large-scale settings is that, if we consider the ambient space $\\mathbb{R}\^d$ in which the statistical manifold $\\mathcal{P}(\\mathcal{X})$ and the sphere $\\mathbb{S}\^{d-1}$ are embedded, the gap between these two diffeomorphic manifolds becomes larger as $d$ increases. Intuitively, this makes transitions from some point to a token (i.e., $\\mathbf{e}\_k$) much farther on the sphere than on the simplex. I am not sure if this is a critical reason why continuous diffusion models struggle to match the performance of discrete diffusion language models.

---

> ### Author Rebuttal · Authors · 2025-07-29
>
> We sincerely thank you for your constructive comments. We appreciate your positive comments that our paper tackles an important topic in current AI research, and the proposed method is elegant and neat. We initially address your comments below:
>
> ---
> >**C1:** While the proposed method is elegant and neat, it is unclear whether there is real benefit for large-scale language modeling.
>
> **R1:** We believe our method offers benefits for large-scale language modeling, both in terms of performance and the inherent advantages of a continuous diffusion framework.
>
> First, superior results on the LM1B dataset, which has a sufficiently large vocabulary set of size 30522, demonstrate strong potential for scaling to large-scale language tasks. Importantly, RDLM can scale to even higher dimensions due to its simulation-free training scheme (Section 4) and dimension splitting technique (Section 5).
>
> Furthermore, our continuous diffusion approach brings additional practical advantages that have been successfully leveraged in other continuous diffusion models. These include controllable generation and compatibility with advanced techniques such as self-conditioning. It also supports scalable and efficient sampling strategies like DPM-Solver, which can further improve inference speed and quality.
>
> ---
> >**C2:** If we consider the ambient space $\mathbf{R}^d$ in which the statistical manifold $P(\mathcal{X})$ and the sphere $\mathbb{S}^{d-1}$ are embedded, the gap between these two diffeomorphic manifolds becomes larger as $d$ increases. Intuitively, this makes transitions from some point to a token (i.e., $e_k$) much farther on the sphere than on the simplex.
>
> **R2:** We would like to clarify that the diffeomorphism between the statistical manifold and the sphere preserves their intrinsic geometric structure, and the notion of gap or distance between points on these manifolds is preserved under this mapping. In particular, the transitions from a point on the statistical manifold to a token $e_k$ on the simplex correspond directly to transitions on the sphere via the diffeomorphism, and the distances do not become inherently larger on the sphere compared to the simplex as $d$ increases.
>
> ---
> >**C3:** Line 135: It says "providing numerous opportunities to correct wrong predictions during the process.", but being continuous can also lead to error accumulation, while discrete transitions hold the possibility of entirely denoising the accumulated error.
>
> **R3:** Our continuous diffusion process is an iterative refinement process, where every intermediate step progressively improves the prediction to the correct target. This iterative nature effectively mitigates error accumulation, which makes the diffusion model a powerful generative model across diverse domains.
>
> In contrast, discrete transitions attempt to reach the correct target in a single jump, which is inherently more challenging and prone to irreversible errors. This is especially critical in masked diffusion models, where incorrect predictions cannot be easily recovered without additional correction mechanisms. Our experimental results consistently show that the continuous diffusion approach outperforms discrete diffusion models, highlighting the advantages of gradual and continuous refinement over discrete jumps.
>
> ---
> >**C4:** Eq. (10): The left-hand side should be $p_{\theta}(X_T)$?
>
> **R4:** In Eq. (10), we parameterize the model $p_{\theta}$ to approximate the transition distribution from current state $X_t$ to final state $e_k$, i.e., $p_{T|t}(e_k|X_t)$. The model takes as input $X_t$ and time $t$, and outputs the the probabilities $p_{T|t}(e_k|X_t)$ for all tokens $e_k$. Therefore, the left-hand side should be $p_{\theta}({X}_t, t)$ as in the paper. We will add a more detailed explanation to clarify this point.

---

> > ### Comment · Reviewer_GUHT · 2025-08-05
> >
> > Thanks for the rebuttal.
> >
> > Regarding the notation in Eq. (10), I understand your intention here, but conventionally we place the random variable whose distribution is being described inside the parentheses of $p(\cdot)$, rather than the conditioning variables or inputs. This differs from the convention used for regular functions, such as the drift $\\eta(X\_t, t)$, where the notation refers to the output and the inputs are placed in parentheses. Perhaps writing it as $p\_{\\theta}(X\_T;X\_t, t)$ or $p\_{\\theta}(X\_T|X\_t, t)$ would make it clearer.

---

> > > ### Author Response · Authors · 2025-08-06
> > > **Thank you for your helpful feedback.**
> > >
> > > We sincerely thank you for your helpful feedback. We agree that using $p_{\theta}(X_T|X_t,t)$ would make Eq. (10) more clear, and will clarify this in the final revision. We hope our response addressed your concerns. We again thank you for your constructive comments.

---

### Note · Authors · 2025-08-15

Dear Area Chairs and reviewers,

We sincerely appreciate your efforts and time in reviewing our paper.

We are grateful that **reviewers have positively highlighted** the originality, theoretical contribution, and empirical strength of our work.

---
>### Novelty and theoretical contribution
- (iVwZ) By grounding diffusion in information geometry, the work unifies discrete and continuous views and offers a principled route to exploit iterative refinement without losing signal at state jumps.
- (FzDR) The methodology of connecting discrete simplex modeling to the generative modeling on the sphere manifold is novel and reasonable
- (myHk) Each component of RDLM (including the theoretical foundation, mixture paths, and training framework) has been discussed and designed in detail.
- (GUHT) proposed method is elegant and neat
- (FzDR) The mathematical derivation is solid

---
>### Superior performance
- ​​(myHk) RDLM and achieve excellent results in text, image, and DNA sequence tasks.
- (iVwZ) Experiments on Text8, LM1B, CIFAR‑10, and promoter‑DNA design show that RDLM is competitive with strong discrete diffusion baselines.

---
We believe we have addressed all the main concerns raised by the reviewers. Below, we summarize our responses for two key points:

- Scalability:

Our superior results on the LM1B dataset, which features a large vocabulary of size 30,522, demonstrate strong potential for scaling to large-scale language tasks. Notably, RDLM can scale to even higher dimensions thanks to its simulation-free training scheme (Section 4) and dimension splitting technique (Section 5).

- Baseline selection:

In our work, we carefully selected state-of-the-art baselines for fair comparison, including continuous and discrete diffusion models as well as autoregressive models, evaluated across diverse tasks. We would like to clarify that models such as LD4LG and TEncDM, which rely on pre-trained language models, are not directly comparable to our approach or the baselines used in our study, as these do not utilize pre-trained models. To accurately assess the core contributions of diffusion modeling techniques, we chose Diffusion-LM, Plaid, and Bayesian Flow Network (BFN) as continuous diffusion model baselines, as they are more directly comparable to our approach under similar training conditions.

---

We sincerely appreciate reviewing our paper providing fair, constructive, and thoughtful feedback.

Yours sincerely,
Authors.

---

### Decision · Program_Chairs · 2025-09-17

**Decision:**

Accept (poster)

**Comment:**

This paper proposes RDLM, a continuous diffusion model for language. The idea is to project the categorical distribution of tokens onto the positive orthant of a sphere, allowing for the use of continuous diffusion processes. The reviews are mixed - the reviewers generally appreciate the novelty of the work but raise concerns about lack of baselines and potential scalability concerns.

Reviewer iVwZ voted strongly for rejection and based that decision on the lack of comparison to related work including LD4LG and TEncDM. After checking the literature, AC stands with the authors that these are not standard baselines in continuous diffusion LMs - as the authors pointed out, these methods rely on pretrained models, which is a different setup from this paper. The authors chose to compare with diffusion LM and Plaid - which is more relevant baselines that do not rely on pretrained models. Therefore, I did not factor this point into my decision.

Given that another key reviewer was on the margin (FzDR pointed out their decision stands in between 3 and 4), I recommend acceptance - the paper's methodological novelty represents a valuable contribution to the field.